# Organic additive release from plastic to seawater is lower under deep-sea conditions

Vincent Fauvelle [1✉], Marc Garel[1], Christian Tamburini [1], David Nerini[1], Javier Castro-Jiménez [2], Natascha Schmidt[1], Andrea Paluselli[3], Armand Fahs[4], Laure Papillon[1], Andy M. Booth [5] & Richard Sempéré [1]

Plastic garbage patches at the ocean surface are symptomatic of a wider pollution affecting the whole marine environment. Sinking of plastic debris increasingly appears to be an important process in the global fate of plastic in the ocean. However, there is insufficient knowledge about the processes affecting plastic distributions and degradation and how this influences the release of additives under varying environmental conditions, especially in deep-sea environments. Here we show that in abiotic conditions increasing hydrostatic pressure inhibits the leaching of the heaviest organic additives such as tris(2-ethylhexyl) phosphate and diisononyl phthalate from polyethylene and polyvinylchloride materials, whereas deep-sea and surface marine prokaryotes promote the release of all targeted additives (phthalates, bisphenols, organophosphate esters). This study provides empirical evidences for more efficient additive release at the ocean surface than in deep seawater, where the major plastic burden is supposed to transit through before reaching the sediment compartment.

[1] Aix-Marseille University, Toulon University, CNRS, IRD, Mediterranean Institute of Oceanography (MIO), Marseille, France. [2] Laboratory of Biogeochemistry of Organic Contaminants (LBCO), IFREMER, Nantes, France. [3] Department of Basic Science, Incheon National University, Incheon, South Korea. [4] Laboratory MAPIEM (EA 4323), University of Toulon, Toulon, France. [5] SINTEF Ocean AS, Environment and New Resources, Trondheim, Norway. ✉email: vincent.fauvelle@ird.fr

Global plastic production exceeded 400 million tons in 2015[1], and it is expected to represent up to 20% of global oil consumption by 2050[2]. A few percent of the plastic produced reaches the global ocean each year[3–6], making the plastic annual flux to the ocean comparable to other major contaminant fluxes (i.e., Tg per year magnitude)[7]. After being released into the marine environment, macroplastic waste is subjected to photo- and biodegradation, in addition to physical aging processes[8], giving rise to plastic debris at the micro-and nanoscale[9,10]. As these degradation processes are very slow, however, the flux of micro- and nanoscale particulates from terrestrial environments is thought to be the major source of marine pollution[11]. Plastic degradation is also considered a significant source of dissolved organic carbon (DOC) release[12–15]. This plastic-sourced DOC includes oligomers at different oxidation levels[13], as well as plastic additives such as phthalates (PAEs), organophosphate esters (OPEs), and bisphenols (BPs)[14,16]. By mass, plastic additives represent 6% of global plastic production[1] and provide polymers crucial characteristics like flexibility or resistance to heat and light degradation.

Most plastic additives that have been detected in coastal and open ocean environments[17–19] are endocrine disruptors[20] and may interact with marine living organisms through (i) release from ingested plastic debris, (ii) assimilation by contact with the chemicals in the dissolved water phase, or (iii) food web transfer[21]. Toxic effects in marine organisms and potential carcinogenic effects in humans have been highlighted for PAEs[22], OPEs[23], and BPs[24], as well as plastic leachate mixtures[25,26]. These compounds are not chemically bound to the polymer and the release of their water-extractable fraction is thus considered to be primarily governed by polymer to water partitioning ($K_{pw}$)[27]. Furthermore, naturally occurring marine prokaryotes have been identified as a crucial parameter in influencing PAE release from plastic to the aqueous phase[14].

A modeling exercise showed that only 1% of the estimated plastic waste is actually floating at the ocean surface[28], suggesting that most of the plastic load is distributed in the ocean interior and very likely at the sediment surface (especially for the denser particles)[29]. This hypothesis is reinforced by evidences from trawling and remotely operated vehicle observations of plastic debris accumulation above the seafloor[30,31], and from elevated plastic additive concentrations at the bottom of the water column in the Yellow Sea and the Mediterranean Sea[17,19,32], which are otherwise two of the most impacted marine areas at the global scale[28]. However, there is insufficient knowledge about the processes affecting plastic distributions and degradation and how this might influence the release of additives under varying environmental conditions. Such knowledge is needed for the whole water column as conditions can be highly contrasted between the surface and the deep-sea in terms of water chemistry, hydrostatic pressure, and prokaryotic content[33].

The current study compares the leaching to seawater of organic additives from selected standardized plastic materials by a natural prokaryote assemblage under surface conditions and high-pressure conditions encountered in the deep-sea (e.g., surface waters and 1000 m depth, i.e., 0.1 vs. 10 MPa of hydrostatic pressure). This work presents investigations into the role of prokaryotes in the degradation of plastic materials and the leaching of organic additives from plastic to deep seawater under high-pressure conditions. Our results show that (i) marine prokaryotes in surface seawater, and to a lesser extent in deep seawater, promote the release of PAEs, OPEs, and BPs, and (ii) high hydrostatic pressure inhibits the leaching of the heaviest and more hydrophobic organic additives such as the PAE diisononyl phthalate (DiNP) and the OPE tris(2-ethylhexyl) phosphate (TEHP). This study provides empirical evidence of a more efficient additive release at the ocean surface than in deep seawater, where the major plastic burden will transit through before reaching the deep seafloor.

## Results

**Additive release in seawater from plastic debris.** Polyethylene (PE) and soft polyvinylchloride (PVC) plastic pellets (3.6–3.7 mm average diameter), which are among the most abundant polymers in the marine environment[11], were added to batches of natural surface and deep seawater and exposed in the dark at 13 °C (Mediterranean deep seawater temperature) for 30 days. For deep seawater, incubations were performed at two different hydrostatic pressures (0.1 and 10 MPa, respectively) using high-pressure bottles (HPBs)[34], abiotic control samples were treated with mercury chloride (HgCl₂). Together with each biotic and abiotic sample treatment, plastic-free controls were deployed under the same operating conditions. At each sampling time, two independent 130-mL bottle samples were sacrificed for chemical and biological analyses. Prokaryotic abundance was measured in each individual sample by flow cytometry. In the absence of a clear information about the additive mix present inside our standard PE or PVC polymer, a total of 25 organic additives were monitored over the duration of the experiment, including 9 PAEs (dimethyl phthalate, DMP; diethyl phthalate, DEP; diisobutyl phthalate, DiBP; di-n-butyl phthalate, DnBP; benzylbutyl phthalate, BzBP; diethylhexyl phthalate, DEHP; di-n-octyl phthalate,DnOP; diisononyl phthalate, DiNP; diisodecyl phthalate, DiDP), 9 OPEs (tripropyl phosphate, TPP; tri-iso-butyl phosphate, TiBP; tri-n-butyl phosphate, TnBP; tris-(2-chloroethyl) phosphate, TCEP; tris-(2-chloro, 1-methylethyl) phosphate, TCPP; tris-(2-chloro-, 1-chloromethylethyl) phosphate, TDCP; triphenyl phosphate, TPhP; 2-ethylhexyl-diphenyl phosphate, EDHPP; TEHP), and 7 BPs (BPA, -AF, -AP, -F, -P, -S, -Z).

Results showed that 5 of the 25 additives monitored were leached from the PE samples during the time course experiment (Table 1 and Supplementary Fig. 1), irrespective of the hydrostatic pressure and biotic conditions, including 2 OPEs (TnBP, TEHP) and 3 PAEs (DMP, DEP, DEHP). In addition, DiNP and BPS were detected in leachates from the PVC samples. For both PE and PVC, most additives were released within the first week of exposure before reaching a zero-flux until the end of the 30-day experiment. An exception was BPS release from PVC, which followed a linear release (i.e., constant flux) over 30 days solely under biotic conditions (Supplementary Fig. 2). The cumulative additive release from PE was always below ppm levels, irrespective of the biotic conditions and hydrostatic pressure applied. BPS release from PVC was below ppm levels for all conditions, whereas DiNP release from PVC was measured in the range 4–88 µg g⁻¹, which is up to two orders of magnitude higher than the cumulative release of all additives observed for PE. In both cases however, the additive release was far below the typical level initially included in the polymers during manufacture (1–10% for PE vs. up to 70% for PVC)[16]. Water chemistry under abiotic conditions (surface vs. deep seawater, treatment B vs. C; Fig. 1 and Supplementary Fig. 1) was not found to play a significant role in influencing in additive release. Two additives with contrasting physico-chemical properties were selected and used as model chemicals in the rest of the study to illustrate the effect of high hydrostatic pressure and prokaryotes on organic additive release: TEHP (OPE, log $K_{ow}$ 9.5, water solubility 0.6 mg L⁻¹) and DMP (PAE, log $K_{ow}$ 1.6, water solubility 4000 mg L⁻¹). This selection was driven by a principal component analysis performed on all additives release data, showing two groups on the second axis (Supplementary Fig. 3).

**Effect of hydrostatic pressure on additive release.** Under abiotic conditions, PE and PVC exposed to atmospheric (0.1 MPa) and high pressure (10 MPa) within HPBs exhibited additive leaching to seawater ranged from several ng to several µg for individual

**Table 1 Organic additive release from polyvinylchloride (PVC) and polyethylene (PE) to seawater under contrasting conditions of hydrostatic pressure and prokaryotic content for 30 days.**

| Polymer | Treatment | A | B | C | D | E | Additive molecular weight (Da) |
|---|---|---|---|---|---|---|---|
| | Seawater source | Surface | Surface | Deep | Deep | Deep | |
| | Pressure (MPa) | 0.1 | 0.1 | 0.1 | 10 | 10 | |
| | Additive\Bacteria | Biotic | Abiotic | Abiotic | Abiotic | Biotic | |
| PE | DMP (ng g$^{-1}$) | 38.0 ± 6.2 | 13.8 ± 1.9 | 14.7 ± 3.3 | 12.9 ± 0.2 | 24.8 ± 5.9 | 194 |
| PE | DEP (ng g$^{-1}$) | 22.8 ± 1.5 | 8.0 ± 0.9 | 7.4 ± 0.7 | 5.9 ± 0.6 | 13.9 ± 2.7 | 222 |
| PE | DEHP (ng g$^{-1}$) | 222 ± 25 | 64.3 ± 27.0 | 33.6 ± 5.9 | 8.9 ± 3.7 | 31.1 ± 0.2 | 390 |
| PE | ΣPAEs (ng g$^{-1}$) | 282.8 | 86.1 | 55.7 | 27.7 | 69.8 | |
| PE | TnBP (ng g$^{-1}$) | 421 ± 24 | 183 ± 12 | 149 ± 10 | 209 ± 14 | 302 ± 19 | 266 |
| PE | TEHP (ng g$^{-1}$) | 34.7 ± 4.2 | 8.3 ± 0.7 | 7.8 ± 0.5 | <1.6 | 2.4 ± 1.0 | 434 |
| PE | ΣOPEs (ng g$^{-1}$) | 455.7 | 191.3 | 156.8 | 209 | 304.4 | |
| PE | Cumulative (ng g$^{-1}$) | 738.5 | 277.4 | 212.5 | 236.7 | 374.2 | |
| PVC | BPS (ng g$^{-1}$) | [0.2 ± 0.1] | 2.5 ± 0.4 | 2.7 ± 0.3 | 3.3 ± 0.4 | [0.2 ± 0.1] | 250 |
| PVC | DiNP (ng g$^{-1}$) | 88,167 ± 23,586 | 17,582 ± 2394 | 18,201 ± 3729 | 4136 ± 629 | 13,745 ± 4230 | 418 |
| PVC | Cumulative (ng g$^{-1}$) | >88,167 | 17,584.5 | 18,203.7 | 4139.3 | >13,745 | |

The values show the maximum additive release at the end of the exposure, when a plateau is observed with first-order kinetic modeling based on 10 or 14 independent data (individual 130 mL samples in duplicate sacrificed at each sampling time for each treatment, ng g$^{-1}$). Each value is reported with ±standard deviation. Brackets contain the values for which a linear regression was applied (i.e., regression slope in ng of additive released per gram of polymer per day). The additives detected were dimethyl phthalate (DMP), diethyl phthalate (DEP), diethylhexyl phthalate (DEHP), diisononyl phthalate (DiNP), tri-n-butyl phosphate (TnBP), tris(2-ethylhexyl) phosphate (TEHP), and bisphenol S (BPS). The two different hydrostatic pressures are equivalent to atmospheric pressure at the surface (0.1 MPa) and 1000 m depth (10 MPa). Deep seawater at 0.1 MPa under biotic conditions was not investigated because it was not relevant for natural deep prokaryote development.

compound per gram of polymer over the 30-day experiment (Table 1, Fig. 1, and Supplementary Fig. 1, treatments C vs. D). The 95% confidence intervals overlapped for the low molecular weight additives such as DMP and DEP during the whole experiment. In contrast, the 95% confidence intervals were separated for higher molecular weight additives such as TEHP, DEHP, and DiNP after 15 days (Supplementary Fig. 1 and Fig. 1), meaning their leaching decreased significantly with increasing hydrostatic pressure.

**Effect of prokaryotes on additive release.** In addition to the abiotic pressure studies, two complementary experiments were performed with surface and deep seawater where the natural prokaryotic assemblage together with the in situ water chemistry was maintained and investigated as potential drivers for organic additive release from PE and PVC to seawater (Table 1 and Figs 1 and 2). Seawater sampling was carried out in late spring, at the time of water mass stratification (see Supplementary Fig. 4), which involves different prokaryotic assemblages between deep and surface seawater[33]. Irrespective of the hydrostatic pressure applied (0.1 or 10 MPa) and the nature of the chemical considered, the presence of prokaryotes adapted to deep-sea conditions and in their original chemical environment significantly increased the amount of additives released from plastic pellets into the dissolved phase (separated 95% confidence intervals; Supplementary Fig. 1 and Fig. 1, except for DEP under deep seawater conditions).

The apparent quantity of additives measured in the dissolved water phase under biotic conditions is the result of two processes: (i) the net flux from plastic, and (ii) waterborne biodegradation. OPEs exhibit short half-lives in phosphorous-limited seawater (2 and 3 days for TEHP and TnBP, respectively)[35], while they are considered persistent in other studies[36]. In the current study, the natural seawaters used are not phosphorous limited and there are no available half-life data in the literature for such conditions[35]. PAEs have half-lives ranging from 26 to 79 days[14], and that of BPS was estimated at 30 days[24]. Given the short duration of the current experiment (30 days), biodegradation is likely to account for a maximum of 50% of the apparent release, except in the case of OPEs where no appropriate biodegradation data are available.

In addition to the 25 targeted compounds, 7 PAE monoester metabolites (MPAEs) were quantified in the exposure samples. Interestingly, two of them (monomethyl phthalate, MMP; and monoethylhexyl phthalate, MEHP) were detected at higher concentrations in the leachates produced under biotic conditions. After 30 days of exposure, MMP was released from PVC in a significantly larger amount under biotic compared to abiotic conditions (146 ± 12 vs. 24 ± 5 ng g$^{-1}$ and 77 ± 17 vs. 22 ± 3 ng g$^{-1}$, for surface and deep seawater, respectively). The release of MMP from PVC could be attributed to biodegradation of DiNP since it was demonstrated that PAEs with longer side chains can be converted to shorter chains by β-oxidation and eventually transesterification[37,38]. A higher concentration of MEHP was also released from PE after 30 days of exposure under biotic compared to abiotic conditions (16 ± 4 ng g$^{-1}$ vs. not detected and 9.7 ± 2.2 ng g$^{-1}$ vs. not detected, for surface and deep seawater, respectively). Again, this is in line with the detection of its precursor DEHP (Table 1). MPAE production mainly results from both photo- and biodegradation[39,40]. As all experiments were performed in the dark, MPAE production cannot be attributed to photodegradation. Their initial presence in the polymer prior to exposure explains their release from plastic under abiotic conditions, whereas PAE biodegradation by bacteria during biotic experiments may explain the higher MPAE production observed under biotic conditions. It is therefore possible to conclude that both surface and deep-sea natural marine prokaryotic assemblages are able to degrade PAEs into MPAEs.

**Effect of plastic leachate on natural prokaryote assemblage.** Systematic prokaryote counting by flow cytometry indicated that the initial filtered seawater (glass fiber filter 0.7 μm) contained low prokaryotic abundance (10$^4$ cells per mL range), corresponding to the free-living fraction of prokaryotes. This increased to 10$^5$ cells per mL after a few days of incubation, for both controls and PE incubations in surface and deep seawater. Figure 2 shows that the addition of PVC to surface or deep seawater decreased the number of prokaryotes by a factor of 4 after 30 days compared to plastic-free controls (0.42 ± 0.09 vs. 1.9 ± 0.06 and 0.54 ± 0.12 × 10$^5$ vs. 2.2 ± 0.22 × 10$^5$ cells mL$^{-1}$ for surface and deep seawater, respectively), with the final number remaining close to the initial value (10$^4$).

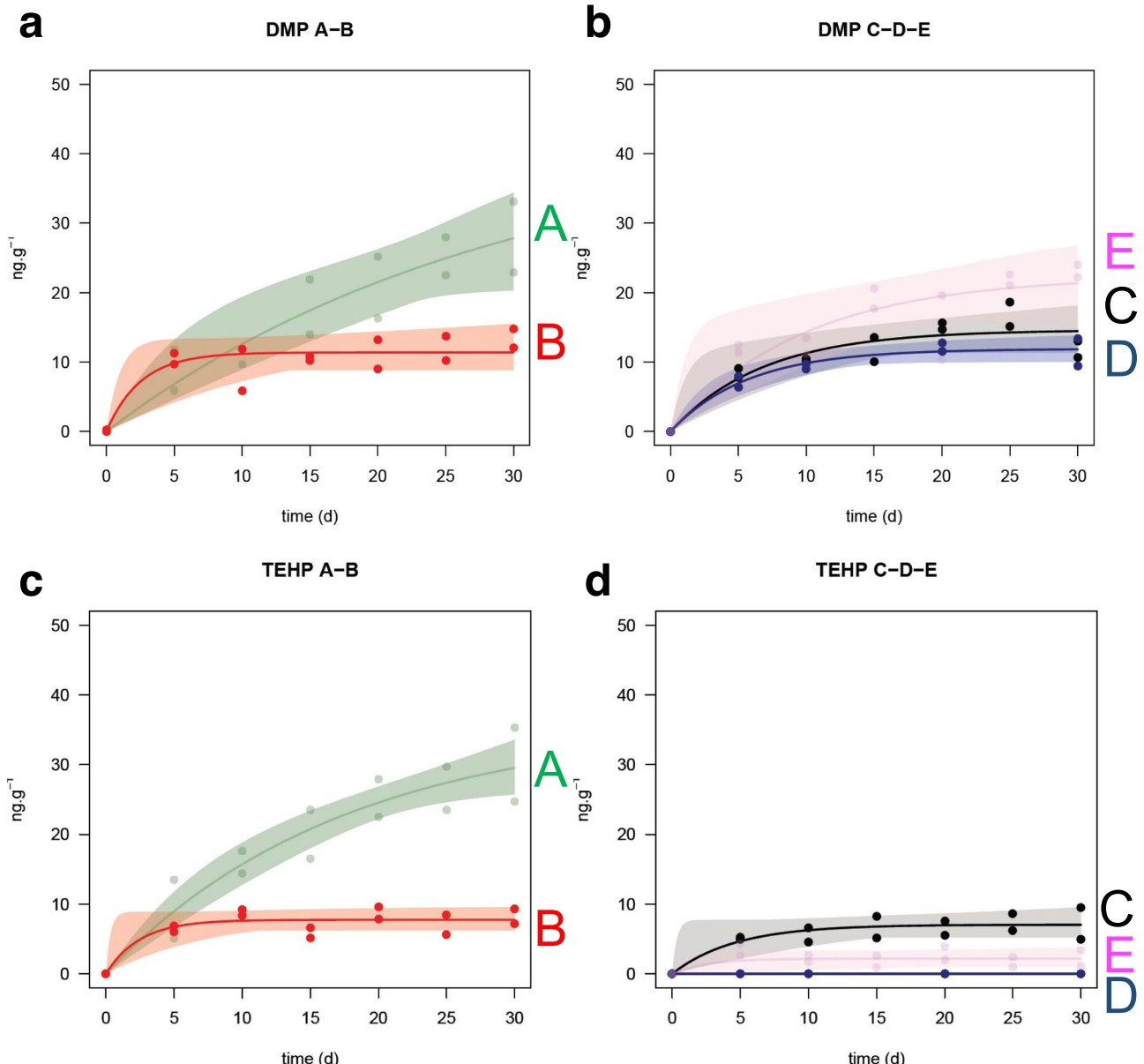

**Fig. 1 Additive release from polyethylene.** DMP (dimethyl phthalate, **a**, **b**) and TEHP (tris(2-ethylhexyl) phosphate, **c**, **d**) leaching and associated 95% confidence intervals from polyethylene over a 30-day exposure under varying hydrostatic pressure and biology. Solid lines are obtained from first-order kinetic fitting. TEHP is representative of more hydrophobic and higher molecular weight compounds and DMP is representative of more hydrophilic and lower molecular weight compounds. Treatments A, B, C, D, and E correspond to those mentioned in Table 1: surface seawater, 0.1 MPa, biotic (A, left column, green); surface seawater, 0.1 MPa, abiotic (B, left column, red); deep seawater, 0.1 MPa, abiotic (C, right column, black); deep seawater, 10 MPa, abiotic (D, right column, blue); deep seawater, 10 MPa, biotic (E, right column, pink).

As there was no measurable elimination of C, N, or P during the PVC incubations (Fig. 3), the decrease in prokaryotic abundance does not appear to be related to a nutrients-limiting process. Rather than decreasing, seawater DOC and dissolved organic phosphorous (DOP) increased significantly in the PVC exposures. As a result, the reduction of prokaryotic abundance in the presence of PVC could be related to the deleterious effect of some of the chemicals constituting the DOC or DOP. Among them, the amount of DiNP (13.7–88.2 $\mu$g g$^{-1}$ of PVC after 30 days of exposure, i.e., 527–3392 $\mu$g L$^{-1}$ under the operating conditions employed; Table 1) and BPS (6 ng g$^{-1}$ of PVC after 30 days of exposure, i.e., 230 ng L$^{-1}$ under the operating conditions employed; Table 1) quantified in the dissolved fraction could be responsible for all or part of the observed limitation in prokaryotes development compared to the control and PE

treatments (Fig. 2). Interestingly, under surface and biotic conditions illustrated in Fig. 3, the release of DiNP (88.2 $\mu$g g$^{-1}$ of PVC, i.e., 65.8 $\mu$g C g$^{-1}$ of PVC; Table 1) accounted for 18% of the measured total DOC release (30.7 $\mu$mol g$^{-1}$, i.e., 368 $\mu$g C g$^{-1}$ of PVC) under the surface and biotic conditions illustrated in Fig. 3. Other non-targeted chemicals such as vinyl chloride monomer or oligomer, PVC nanoparticles (i.e., particle size below the 0.7 $\mu$m filter cutoff), or non-targeted additives could also have promoted the decrease in prokaryotic abundance.

## Discussion

Laboratory exposure of PE and PVC pellets under contrasting conditions of hydrostatic pressure and heterotrophic prokaryotes

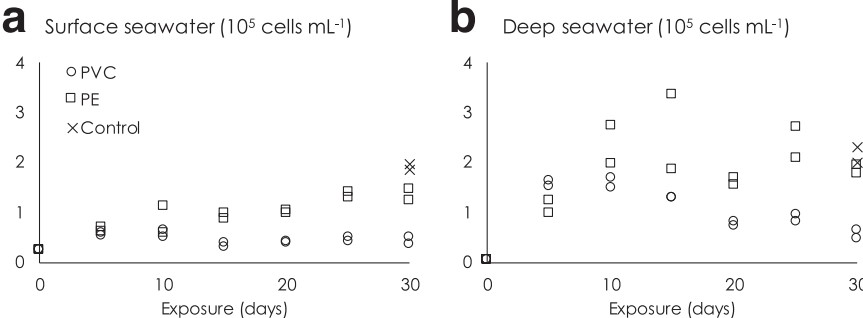

**Fig. 2 Prokaryotic content in seawater.** Changes in prokaryotic abundance ($10^5$ cells mL$^{-1}$) over time in the presence of polyethylene (PE, squares) and polyvinylchloride (PVC, circles) pellets under surface (**a**) and deep seawater conditions (**b**). Controls are seawater incubated with no polymer for 30 days under biotic conditions (crosses).

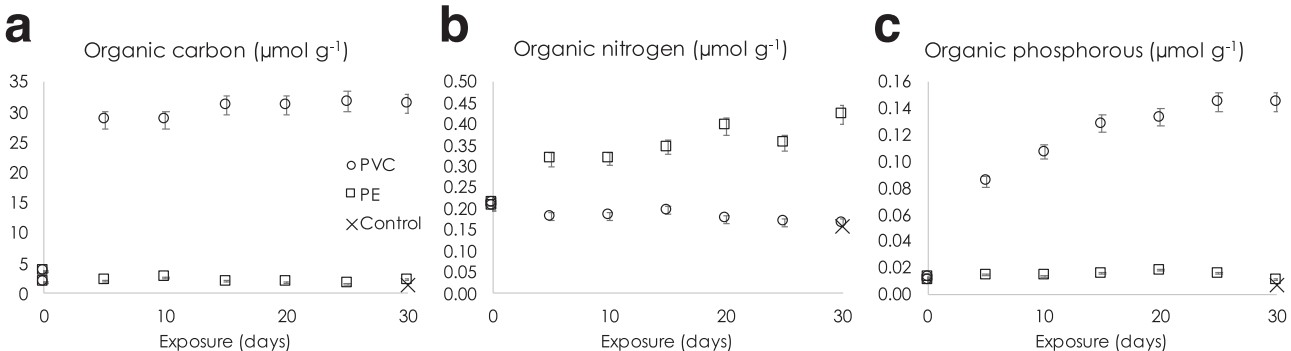

**Fig. 3 Organic carbon, nitrogen, and phosphorous in seawater.** Dissolved organic carbon (DOC, **a**), phosphorous (DOP, **b**), and nitrogen (DON, **c**) released into seawater per gram of polyvinylchloride (PVC) or polyethylene (PE) over time during the course of the experiment under biotic conditions at atmospheric pressure 0.1 MPa. Controls consist of seawater incubated with no polymer pellets for 30 days under biotic conditions. The diameter of the PVC and PE pellets is in the 3–5 mm range. Error bars show standard deviation ($n = 3$).

content revealed systematic leaching of a range of organic additives. Leaching occurred within the first week before reaching a zero-flux after 1 month of exposure, except for BPS. The study did not aim to provide a comprehensive characterization of the mixture of organic additives present in the polymers studied. However, based on the list of 25 target additives, it was evident that (i) PVC released two orders of magnitude more additives than PE, (ii) DiNP was the main compounds measured in PVC and that OPEs were the predominant chemical family leached from PE, and (iii) PE released a wider diversity of additives. It should be noted that the PE and PVC test materials employed in the current study represent individual examples of these plastic materials and should not be considered representative of all PE and PVC materials and their varying additive chemical profiles.

Otherwise, the results showed that high hydrostatic pressure (i.e., 10 MPa) simulating deep seawater conditions diminished the release of the more hydrophobic additives (e.g., DiNP, DEHP, TEHP), whereas it had no effect on the more hydrophilic substances detected. This last finding suggests that high hydrostatic pressure, and therefore greater water depth, limits leaching of the heaviest and more hydrophobic organic additives (e.g., log $K_{ow}$ values of DEHP and TEHP are 7.6 and 9.5, respectively). This observation may be explained by (i) an increase of $K_{pw}$ potentially induced by the high hydrostatic pressure, or (ii) a reduction of the water accessible polymer surface microlayer thickness due to physical constraints reducing the pseudo-pore size and polymer free volume (i.e., pseudo porosity) implying eventually a reduction of additive diffusion inside the polymer. The present dataset however does not permit distinguishing between these two hypotheses.

The presence of heterotrophic prokaryotes in their original chemical environment was also found to promote additive release, especially the surface seawater assemblage. While additive leaching under pressurized conditions has never been reported for plastic materials, our results are in line with a previous study[14] indicating favorable release of additives at atmospheric pressure (0.1 MPa) in the presence of prokaryotes. Polymer oxidation by prokaryotes increases the surface hydrophilicity[8] and lowers the $K_{pw}$, which could support this finding. A thicker water accessible polymer layer or a higher specific surface area caused by prokaryotic polymer degradation may also explain these results[14]. When considering only the effects of prokaryotic content and high hydrostatic pressure, these results strongly support that additive release and biodegradation of plastic is more effective at the ocean surface than it is in the deep-sea. In addition, the role of UV exposure from sunlight (not investigated here) drastically increases the degradation capacity of the ocean surface compartment compared to the deep-sea compartment[13,14,41].

The cumulative release of target plastic additives in surface and deep seawaters was found to range from 212 to 738 ng g$^{-1}$ for PE and from 4.1 to 88.2 µg g$^{-1}$ for PVC (Table 1), depending on the specific exposure conditions of high hydrostatic pressure and prokaryotic content, being in the same order of magnitude as those measured in a previous study[14,27]. Considering the estimated levels of plastic discharges to the global ocean proposed by Jambeck et al.[4], together with the polymer manufacturer's distribution mentioned by Geyer et al.[1], it is possible to propose tentative additive chemical release estimates of 2.3–132 tons per year from PVC, and 0.4–3.4 tons per year from PE, occurring within the first week of their introduction to the ocean. It is

important to note that these estimates only include the 25 target additives measured (7 actually detected) in the current study. It is very likely that other non-targeted compounds will be leached from PE and PVC, as well as other plastic materials, meaning the total quantity of additive chemicals released to the oceans is much higher. These results, which are consistent with the available but sparse literature on the topic[27], are 2–3 orders of magnitude lower than the 6% additive content (of which 50% is organic chemicals[42]) mentioned for the global plastic manufacturing data[1]. This suggests that over the initial 30-day period after release to the oceans, most of the plastic additive burden is not readily water accessible from mm-size plastic particles. Instead, additives may be leached as a result of plastic particle degradation, which leads to a long-term diffuse release of additives, and potential long-range transport toward remote marine areas and global sink regions (e.g., the Arctic, deep seawater and sediments, offshore surface waters)[5,18,30,43,44].

Recent investigations suggest that only 1% of the estimated plastic waste entering the oceans is actually floating at the surface[28]. This is most likely due to one or a combination of the following processes: (i) almost complete mineralization of plastic debris, (ii) fragmentation to particle size below the cutoff threshold of the nets used for generating the modeled data (typically 330 μm, i.e., small microplastics and nanoplastics), (iii) migration through the water column due to lower buoyancy of the smallest particles (i.e., <1000 μm), and (iv) sinking of plastic debris toward the seafloor and sediment close to sources when their density is higher than 1, or far from sources when polymer density is lower than 1, after biological colonization and/or heavy oxidation processes that increase particle density[10,44–48]. Sinking increasingly appears to be an important process in the global fate of plastic in the ocean[30,46]. The results of the current study show a two-fold reduction in additive release from PE and PVC in deep seawater compared to surface conditions during the first month of exposure (i.e., high hydrostatic pressure and natural prokaryotic assemblage, without taking into account the effect of UV irradiation). This implies that both plastic and any associated additives will be more persistent in the deep-sea environment compartment. A similar pattern of higher degradation efficiency has already been observed in the euphotic and twilight zones for particulate organic matter (POC)[49–51]. Although far from being negligible in the current study, the additive chemical release observed under deep-sea conditions is associated with the activity of a free-living deep seawater-derived prokaryotic assemblage as initial inoculum. Particulate attached prokaryotes may be capable of dissolving POC more efficiently by enzymatic processes, providing dissolved organic matter for free-living prokaryotes[33]. Nevertheless, the present work focuses on the water column and excludes the sediment compartment, which is very likely the ultimate sink for sinking plastic particles.

Seabed sediment generally exhibits high microbial activity on its oxic surface layer, which might promote plastic aging and subsequent additive release. In deeper sediment layers, diagenetic processes may also operate at slower rates. However, plastic debris remobilization is likely to happen at any stage after initially reaching the sediment surface. As shown by the current study, the action of prokaryotes appears to be the main driver for additive release in deep-sea environments. Gewert et al.[13,41] highlight that plastic aging appears to occur in two distinct stages, first via abiotic processes (where UV-catalyzed oxidation is predominant), and second via biodegradation (mainly driven by bacteria). The combination of initial surface ocean exposure conditions (UV-rich) followed by long-term sediment exposure conditions (prokaryotes activity) during the plastic debris lifecycle likely promotes the degradation of plastic in the marine environment.

In addition to influencing additive chemical release from PE and PVC, the presence of prokaryotes also resulted in the production of additive chemical transformation products (e.g.,

MPAEs). These degradation products are more hydrophilic than the parent compounds and therefore have an increased ability to move through the sediment and water column. This is primarily because their diffusion coefficient in water is higher than their precursors, where typical values[52] are $10^{-8}$ m$^2$ s$^{-1}$ for most hydrophilic organic compounds (e.g., MEHP) compared to $10^{-10}$ m$^2$ s$^{-1}$ for hydrophobic compounds such as DEHP. Furthermore, their affinity for sediment particles is lower (e.g., organic carbon interaction).

The role of prokaryotes in plastic degradation could also be influenced by their growth limitation induced by plastic debris, both for surface and deep communities, as shown for PVC in Fig. 2. It is possible that the leached DOC or DOP reach locally high concentrations, especially in the sediment where slow molecular diffusion through pore water is the main process involved in the migration of chemicals. As a result, the release of a cocktail of substances, including additives, oligomers, and polymer-based nanoparticles, might be responsible for an overall growth limitation of prokaryotes over time, as highlighted otherwise[53]. Our results suggest that the accumulation of plastic debris at the bottom of the water column can be expected to represent a slow but continuous source of additive chemicals over much longer periods of time compared to surface waters. While higher additive release rates might occur in surface waters, exposure levels to deep-sea marine organisms could be greater overall due to the higher concentration of accumulated plastic present in this compartment.

## Methods

**Seawater sampling and pretreatment.** Bulk deep seawater (24 L) was collected at 1000 m from the Cassidaigne Canyon (43.068223 °N, 5.468057 °E, 15 km from Marseille coastline) using two 12-L GO-FLO© GENERAL OCEANICS bottles, previously rinsed with 1% HCl and ultrapure water to avoid contamination. Samples were collected on June 6, 2018 from the R/V Antedon II and filtered through a 0.7 μm GF/F the same day. The temperature and salinity profiles are provided in Supplementary Fig. 4. Bulk surface seawater was collected and filtered following the same protocol at 0.5 m depth in Marseille Bay (43.273624 °N, 5.347348 °E) on November 23, 2018. Seawater was transferred to precombusted (450 °C, 5 h) 130-mL glass bottles together with appropriate treatment (i.e., addition of plastic pellets and HgCl$_2$). Opened caps protected by a polytetra-fluoroethylene (PTFE, 1 mm thickness) septum were used to transfer the hydrostatic pressure to the bottle. A preliminary experiment (same bottle-caps-PTFE septum, 130 mL ultrapure water NaCl 35 PSU spiked with 10 ng of target additives, 7-day incubation, $n = 3$) showed no measurable losses of the target contaminants and therefore no adsorption of additives to the PTFE septum or bottle walls. Deep seawater and surface seawater had the following characteristics, respectively: salinity = 38.5 and 37.9, DOC = 1.3 and 7.2 mg C L$^{-1}$.

**Experimental design.** Reference plastic materials were provided by CARAT GmbH (Bocholt, Germany). The PE consisted of recycled low-density PE pellets (PTX131, density 0.955 g mL$^{-1}$, average diameter 3.7 mm, surface area 17 cm$^2$ g$^{-1}$), whereas the PVC consisted of pristine soft PVC pellets (PTX500, density > 1.1 g mL$^{-1}$, average diameter 3.6 mm, surface area 13 cm$^2$ g$^{-1}$). The release kinetics of the organic plastic additives (PAEs and their primary MPAEs, OPEs, BPs) from the two types of plastic (5.0 ± 0.1 g sample$^{-1}$ of PE or PVC) were studied under laboratory conditions for 1 month in 130 mL of filtered seawater. In these studies, varying environmental conditions of (i) seawater location (surface vs. deep), (ii) hydrostatic pressure (0.1 or 10 MPa), and (iii) prokaryotic activity (sterilized or not with HgCl$_2$, 10 mg L$^{-1}$ final concentration) were investigated. All experiments were performed in the dark at 13 °C in a temperature-controlled laboratory and each condition was deployed in duplicate. The kinetic parameters shown in Table 1 were therefore determined on 10 or 14 datapoints (corresponding to 5 or 7 time points in duplicates). Controls with no polymer were analyzed at $t_0$ (biotic, 0.1 MPa) and $t_f$ (biotic 0.1 MPa, biotic 10 MPa, abiotic 0.1 MPa, abiotic 10 MPa). At each sampling time, two 130-mL bottles per treatment were sacrificed, and the total volume was distributed for analyses in the following manner: 2 mL used for microbiological analysis, with 10 mL used for DOC analysis, 48 mL used for N and P analysis, and the remaining 70 mL kept at −20 °C until analyzed for the plastic additive content.

**Hyperbaric setup.** Selected samples were set at 10 MPa using HPBs. HPBs are 500-mL APX4 stainless steel cylinders (75 mm OD, 58 mm ID, and 505 mm total length) constructed by Metro-Mesures (Mennecy, France). Viton® O-rings are used

to ensure that the system is pressure-tight. Each HPB contained two 130-mL glass bottles sealed with a PTFE septum cap, and was subsequently filled with deionized water to pass on the pressure. Pressure was transferred inside the glass bottle samples via a PTFE septum (see "Seawater sampling and pretreatment" section). The screwed bottom end-cap of the HPB was connected, via a 1/8" stainless steel tube, to the piloted pressure generator to apply hydrostatic pressure inside each HPBs at 0.2 MPa s$^{-1}$ [51].

**Organic additive analysis in seawater**. The target list of 25 parent additive chemicals and 7 transformation products comprised a suite of PAEs, PAEs monoesters (MPAEs), OPEs, and BPs: mono and dimethyl phthalate (MMP and DMP), mono and diethyl phthalate (MEP and DEP), mono and diisobutyl phthalate (MiBP and DiBP), mono and di-n-butyl phthalate (MnBP and DnBP), monobenzyl and benzylbutyl phthalate (MBzP and BzBP), mono and diethylhexyl phthalate (MEHP and DEHP), mono and di-n-octyl phthalate (MnOP and DnOP), DiNP, tripropyl phosphate (TPP), tri-iso-buthyl phosphate (TiBP), tri-n-butyl phosphate (TnBP), tris-(2-chloroethyl) phosphate (TCEP), tris-(2-chloro, 1-methylethyl) phosphate (TCPP), tris-(2-chloro-, 1-chloromethylethyl) phosphate (TDCP), triphenyl phosphate (TPhP), 2- ethylhexyl-diphenyl phosphate (EDHPP), TEHP, BPA, and its analogs BPAF, BPAP, BPF, BPP, BPS, BPZ. Analysis of all chemicals in seawater was adapted from Fauvelle et al.[54] with slight modifications: 70 mL seawater sample volume, PAE monoesters were included in the target list and analyzed from the second elution fraction by liquid chromatography coupled to HRMS. DiNP was also added to the target list and analyzed in the first fraction by gas chromatography coupled to tandem mass spectrometry, after first being identified in the PVC leachates by analyses performed in a full-scan mode. PAEs monoester standards were provided by Cluzeau Info Labo, France. QA/QC consisted of (i) instrumental blanks that were always < limits of quantification (LOQ), (ii) procedural blanks ($n = 18$) that were always < LOQs, except for DEHP that was quantified between 2.7 and 8.5 ng L$^{-1}$ (these values were still ten times lower than concentrations measured in real samples, results were blank corrected for this compound), (iii) spiked water (100 ng sample$^{-1}$, $n = 3$, recoveries comprised between 63 and 107%), and (iv) biotic and abiotic final controls without polymers (no significant losses observed between $t_0$ and $t_f$ whatever the conditions applied).

**Prokaryotic abundance**. For the heterotrophic prokaryote determination, seawater aliquots were analyzed by using the flow cytometry core facility PRECYM of the Mediterranean Institute of Oceanology (http://precym.mio.osupytheas.fr). Immediately after sampling, the samples were thawed at room temperature and stained using SYBR Green II (Molecular Probes®). Analyzes were performed on a FACS-Calibur flow cytometer (BD Biosciences®) equipped with an air-cooled argon laser (488 nm) and a red diode (633 nm)[55]. A figure showing data gating and sorting strategy is available in Supplementary Fig. 5.

**Dissolved organic carbon nitrogen and phosphorous analysis**. A 10 mL aliquot of each sample was filtered through a precombusted 0.7 µm GF/F and transferred to precombusted PTFE-capped glass vials, acidified with H$_3$PO$_4$. DOC concentrations were then measured using a Shimadzu TOC-5000 carbon analyzer[56]. DON and DOP were determined by subtracting the dissolved inorganic N and P concentration (determined using an Autoanalyzer III Seal Bran Luebbe) from the total dissolved N and P concentrations determined using a wet-oxidation procedure described by Raimbault et al.[57].

**Statistical analyses: strategy and approach**. A first-order reaction equation is built to represent the release kinetics of additives from plastic particles to the water phase. Let $X$(ngg$^{-1}$) be a chemical quantity released in a given medium. The dynamics of $X$ is given with:

$$X(t) = a\left(1 - \exp(-bt)\right) \quad (1)$$

where $a$ and $b$ are parameters that must be estimated from the data. Note that $X(t)$ is a strictly concave increasing function that goes to $a$ (ng g$^{-1}$) as time $t$ (d) goes to infinity and where $b$ (ng g$^{-1}$d$^{-1}$) is the rate at which $X(t)$ goes to its plateau. Starting with experimental data $(t_i, x_i)$, $i = 1, \cdots, n$, parameters $a$ and $b$ are estimated by non-linear regression (Gauss–Newton descent-gradient algorithm)[58] when minimizing the cost function:

$$\text{SSE}(a, b) = \sum_{i=1}^{n} \left[X(t_i) - x_i\right]^2 \quad (2)$$

which measures the distance from the data to the model, where $X(t_i)$ is the value of the model at time $t_i$. Initial conditions for parameter estimation are chosen empirically so that the first fit is reasonable. Once estimations $\left(\hat{a}, \hat{b}\right)$ have been found, a confidence region can be designed as a part of the parameter space for which boundaries are defined with:

$$\left\{(a, b) : \text{SSE}(a, b) \le \left(1 + \frac{p}{n-p}F_{p, n-p}^{\alpha}\right)\text{SSE}\left(\hat{a}, \hat{b}\right)\right\} \quad (3)$$

where $p$ is the number of parameters, $n$ is the number of observations, and $F_{p, n-p}^{\alpha}$ is the α-order quantile of the Fisher–Snedecor distribution with $(p, n - p)$ degrees of freedom. For a large number of observations ($n$), this region will have the required asymptotic confidence level $100(1 - \alpha)\%$ (see Seber and Wild for more details)[58] and is also reliable for finite $n$. The kinetics curve can be estimated using pairwise observations $(a, b)$ randomly drawn inside the confidence region. This makes the confidence bands such as those presented in Fig. 1 and Supplementary Fig. 1. In this way, it is possible to compare dynamics between compounds or treatments all along the time course with statistical significance. The more the dynamics are different, the less their 95% confidence intervals overlap.

**Reporting summary**. Further information on research design is available in the Nature Research Reporting Summary linked to this article.

## Data availability
The authors declare that the data supporting the findings of this study are available within the article.

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

## Acknowledgements

The authors are grateful to Christian Grenz and Deny Malengros from SAM-MIO platform, and the crew and captain of the R/V Antedon II for operation at sea. The authors deeply thank PRECYM-MIO platform (Morgane Didry, Aude Barani, and Gérald Grégori) for flow cytometry analysis. France Van-Wambeke is acknowledged for support in microbiology, Patrick Raimbault for nutrient measurements, the "Culture Expérimentale" platform for sample incubation, and Benjamin Oursel for DOC analysis. This work has been conducted as part of the European project "PLASTOX" (Direct and indirect ecotoxicological impacts of microplastics on marine organisms), under the framework of the Joint Programming Initiative – Healthy and Productive Seas and Oceans (JPI Oceans). This study was financially supported by the Agence de l'Eau-CAREMED and Blue-POLUT projects and by the Research Council of Norway (grant agreement no. 257479). The project leading to this publication has also received funding from European FEDER Fund under project 1166-39417.

## Author contributions

V. F., M. G., C. T., and R. S. designed the experiment; V. F., N. S., J. C.-J., A. F., and L. P. performed the experimental work; A. P., A. M. B., V. F., R. S., and D. N. did the interpretation of the data; V. F. and R. S. wrote a first version of the manuscript, and all co-authors made comments and approved the final version.

## Competing interests

The authors declare no competing interests.
