## [Peer Review File · Nature Communications]

Reviewers' comments:

Reviewer #1 (Remarks to the Author):

Summary

The given paper claims on the leaching process of plastic additives. In this context preproduction pellets of two representative, pristine polymers, polyethylene (PE) and plasticized polyvinylchloride (PVC) were treated in a seawater environment for 30 days. Representative for surface and deep sea conditions the experiments were carried out at 0.1 MPa with surface and 10 MPa with deep sea water, respectively. Here the leaching processes were performed in the dark with seawaters natural prokaryote assemblage and abiotic seawater (HgII+ treatment). The four different conditions were compared focusing the influence of pressure and presence of prokaryotes on the leaching process. The leachates were analyzed for 23 additives, 7 phthalates (PAEs) and 9 organophosphate esters using GC/MS in SIM mode, and 7 bisphenols using LC/QTOF. As a result 3 phthalates and 2 organophosphates in PE leachates (in sum between 212 and 739 ng per g polymer), and BPS as only additive in PVC leachates (in sum 0.2 to 3.3 ng per g polymer) were identified and quantified. This led to the conclusion, that cumulative release of additives below ppm levels is orders of magnitudes lower than percent levels included during production. Representative for the PE leachate Tris(2-ethylhexyl) phosphate (TEHP) with high log Kow and dimethyl phthalate (DMP) with of high and low log Kow were chosen to describe effects of pressure and prokaryotes. Additives were leached less at higher pressure conditions. This effect was particularly expressed for TEHP (=high Log Kow). For cumulative data the factor was around 2 suggesting a reduced additive release under increased water pressure. Additives were leached to a higher extent at biotic conditions at same pressure conditions, and more expressed under surface conditions in particular regarding TEHP. This draws the conclusion to an increased additive release at the presence of bacteria. Additionally 7 phthalate monoesters (MPAEs) metabolites (only two are specified, five are missing L 170/L393) were analyzed with LC/QTOF as well. Of those two were detected in the PVC leachate with highest concentration at biotic and surface conditions (146 ng per g polymer after 30 days) point to not targeted and therefore not analyzed PAEs. One MPAE in the PE biotic conditions leachates were detected as well (16 and 9 ng per g polymer). Their presence was led back to microbial biodegradation. The leachate of PVC led to decreased bacteria counts compared to PE leachate after 30 days and was led back to toxic effects of leached non-targeted OPEs or PAEs. All results were discussed, a model calculation of globally released additives based on the data and finally an outlook were given.

Remarks

The idea to perform additive leaching experiments under increased pressure (deep sea conditions) is interesting and so far not published. Same holds true for the parallel studies of biotic/abiotic conditions and its influence on additive leaching. Regarding the seafloor as a relevant sink for plastic litter and degradation products the analytical setup and achieved results are of general relevance. The paper is written in fluent and well readable English.

The observed results and trends are stimulating and worth to be followed but, nevertheless, very limited and not convincing on the presented level. Therefore the overall conclusions are not convincing and even partly misleading at the present stage.

The major point of critique regarding the study is the fact, that the authors did not characterize the pristine polymers used in the experiments with regard on their additive content in advance. From my point of view this is fundamental for this kind of study in particular when working with a model system. The authors don't seem to have reviewed details about usual additive contents of specific polymers rather than on a very general view.

PE is a polymer with relatively low additive content (1-2%) and rarely plasticized. On the other hand soft PVC contains approximately 30% of plasticizers (33% (density 1.22), 44% (density 1.17), PVC-Handbook, Wilkes et al. 2005) and the stabilizer content (e.g. BPA or its substitute BPS) is << 1%. Most common plasticizers used for PVC are diisooalkyl phthalates (e.g. octyl, nonyl, decyl), technical mixtures of isomers, no easy to analyze but definitely to be included in such kind of study.

PVC is easily solvable in e.g. THF. It is no problem at all to identify and even quantify its containing

additives after polymer precipitation using GC/MS (for the here relevant PAEs and related plasticizers) and of course in full scan mode. This is very easy step and obligate for an analytical question as it is asked by the authors. Working directly on a targeted (SIM)-level and analyzing the leachates only makes the analysis set up "blind" for other compounds and consequently might led to misleading and partial results. Same holds true for PE, even if it is more difficult to get into solution, a severe leaching with appropriate solvent would have given valuable insights.

Having this in mind it is highly doubtful that PE releases more additives like plasticizers than PVC (Table 1), and PVC leachate contains BPS, only. Consequently, Table 1 represents an incomplete picture of the analytical setup and conclusions like

L 122-123 "The cumulative release of additives from polymers was always below ppm levels..."
or

L 227-228 "PE released more organic additives than PVC, both in quantity and diversity..."
are not supported.

The given, relativizing argumentation in L228-230 "Although no measurable leaching of PAEs was observed from PVC, it is possible that non-targeted PAEs or OPEs may have leached from this polymer", even if it is true from the data sets point of view, doesn't change the fact that the given results reflect a distorted picture of the leachate. The additional argumentation (L 230 – 234) is not an excuse at all for the missing investigation and characterization of the additives representing at least 1/3 of the used soft PVC pellets by weight! Even though the authors know their leak of knowledge regarding the PVC, they underestimate its impact by far and make any further conclusion doubtful.

The presence and discussion of PAEs metabolites (L170ff) where those of PVC exceed those of PE by far underline the above critique even the argument of non-targeted PAEs is stressed again.

L 257ff General calculations on overall additive releases into the ocean founded on the given database seem to be very vague and doubtful in particular regarding the PVC share. Paluselli et al. 2019 analyzed only a very restricted number of PAEs and not the leachate in general; their study has the same weak point and should be handled with caution as well regarding some of the given generalized results, disregarding some very interesting and fundamental other aspects presented there. Literature data cited by the authors as well reported relevant higher additive leaching for PVC (upper µg-range per g polymer, Suhrhoff and Scholz-Böttcher, 2016) in model systems. The argument that non targeted additives are missing should consequently exclude respective calculations.

L 323ff The conversion of the observed cell number decrease into a toxic effect of plastic debris on natural bacterial assemblages is a fair hypothesis but should not be dramatically overdrawn as written in L327-329 "As a result, the release of a cocktail of substances, including additives, oligomers, and polymer-based nanoparticles might be responsible for an overall decrease in bacteria content over time due to toxic effects". Is there any indication, that there is such an effect (as implied by the sentence already) and is it realistic that (the very low) plastic abundance in sediment release a toxic leachate level by the state of knowledge so far? Here, a more hypothetical and scientific grounded argumentation would be preferred.

At the presented level the data do not allow that high level of generalization regarding the possible impact of additive leaching in deep sea environment (L 278 ff). The same holds true regarding the conclusion that "biodegradation is the main driver for plastic degradation and additive release in deep-sea environments" (L302 ff). Regarding the presented data no clear relation to a polymer degradation process is documented. The discussion is suggested to be performed more result-related, more hypothetical and less general.

The analytical equipment used by the authors would allow a consistent and comprehensive experimental data set. A repetition of the experiment with a profound prior analysis of the additive content of the "representatively" used polymers and a subsequent related analysis of the leachates first in full scan MS mode and successively on an adapted set of targeted additives will considerably improve the quality of the results. Subsequently derived conclusions are than supposed to be much more profound and convincing for superordinate conclusions.

In the current state I would recommend a rejection of the paper, according to the argumentation above.

If the leachates are still available they should be reanalyzed targeting those additives identified in the PVC/PE used for the experiments in an obligate first step. Analysis of leachates should be performed in full scan and SIM mode as well with regard to further degradation products (e.g. related MPAs). An accordingly revised manuscript should give a much more convincing picture of the experimental setup and derived conclusions. In this case a resubmission should be offered.

Reviewer #2 (Remarks to the Author):

Recommendation: Accept with minor revisions

General Comments

This paper presents results of lab experiments that measured the amounts of phthalates, organophosphates and bisphenols leached from polyethylene and PVC at different pressures and in the presence and absence of prokaryotes. The results are interesting in that they provide some clues about how the persistence of plastic materials is affected by sinking in the ocean.

The paper is well written and clearly presented. I support its publication after the authors consider my specific comments and suggestions below.

Specific Comments

Line 41: Another study in addition to (13) that shows polymer chain scission is a source of DOC is this recent paper by Colin Ward & Chris Reddy (<https://pubs.acs.org/doi/abs/10.1021/acs.estlett.9b00532>)

Line 46: "Most additives that have been detected in the ocean are endocrine disruptors... food web transfer toward higher trophic level organisms". Looking at the references for this sentence, I think the authors are conflating different issues and lumping them together in a way that will mislead readers and probably overstates the (known) problem. The statement that "most are EDCs" seems to be referring to phthalate esters and bisphenol-A. But, phthalates and bisphenol-A are not bioaccumulative, and although they are transferred to higher trophic levels they undergo "trophic dilution" and are thus not considered a high priority due to B (see for example <https://pubs.acs.org/doi/full/10.1021/es034745r> and <https://doi.org/10.1016/j.chemosphere.2015.08.059>).

Line 78: "This study provides the first empirical evidence of a more efficient plastic debris degradation and subsequent additive release at the surface than in deep-sea..." I think the wording here is a little too loose. I think it is plausible that this is the first study showing more efficient additive release at the surface. But I am quite confident there are studies out there that provide empirical evidence that degradation of plastic debris is more efficient at the surface (due to UV light exposure if nothing else, as the authors themselves write on lines 250-252!).

Line 81: The Results section is very nicely organized and presented!

Figure 2: As a reader, I want to make visual comparisons between the abiotic and biotic treatments, but also between the two pressures. So the data in the upper and lower panels for TEHP and DMP should be presented on the same vertical scale. This will not be a problem at all for DMP. For TEHP the data at 10 MPa will be close to the bottom of the plot, but I think that is OK since there is no trend in the data anyway...

Figure 3: As above, I would like to see these two panels presented with the same vertical scale so I can instantly visually compare the surface seawater data with the deep seawater data.

Line 297–307: This paragraph is a nice synthesis of when and where degradation of plastic polymers can happen in aquatic systems.

Line 315: “Monitoring...” This is an interesting idea, but my first reaction is that it will not work because there are so many sources of phthalates to the environment. I somehow doubt that leaching from plastic in-situ in the environment is a dominant source of phthalates in water such that the concentrations (or even the ratios of concentrations of monoesters to diesters) could be used to map plastic pollution. Maybe the authors could do some back-of-the-envelope calculations based on their experiments and compare them to phthalate concentrations reported in the environment to check if their suggestion is reasonable.

Line 511: There is an amusing typo in the title of Ref. 27 (Fl Oating).

Reviewer #3 (Remarks to the Author):

General Comments

This manuscript describes a study designed to compare the leaching of organic additives from microplastic to marine water at the ocean surface as compared to the deep ocean. Two types of plastic beads (polyethylene and polyvinylchloride) were incubated with sea water collected from the ocean surface (0.5 m) or at 1000 m depth. Samples were incubated under two pressure scenarios, 0.1 and 10MPa, for 30 days and the release of 23 organic additives was quantified. The manuscript is well written and clear and the results are presented clearly and effectively. The study produced some interesting results, including that the leaching of heavy molecular weight organic additives was lower under high pressure, and that the presence of microbes increased the release of organic additives.

My main concerns about this manuscript relate to experimental design, replication, and application of statistics. The study is attempting to address several independent variables (pressure, seawater source, presence vs. absence of live bacteria) but the design is not fully crossed, e.g. there were no incubations of surface water at high pressure and no biotic incubations of deep water at low pressure. A fully crossed design would have enabled a more rigorous statistical analysis of their data, e.g. using analysis of variance, which would have assessed individual effects of independent variables and interactions. As it stands the statistical analysis of their data is limited to a few haphazardly applied t-tests which are only mentioned parenthetically in the Results section. Moreover, the degree of replication included in the study is unclear. Table 1 states that their $n = 10$ or 14 , but in line 364 it states "each sample was deployed in duplicate". If the degree of replication was duplicated for each treatment, this is not adequate for a rigorous statistical analysis. This problem is exacerbated by the fact that there is no mention of statistics in the Methods section. The authors need to explain more clearly how their data were statistically analyzed and to address the limitations of their study design.

The study assumes that the prokaryotic communities from 0.1 m and 1000 m are different in taxonomic composition. This is almost certainly true, but it would have been nice to have seen this documented, for example via 16S amplicon sequencing.

The study does not present, account for, or discuss the possible effects of physical or chemical differences (e.g. pH, osmotic concentration, dissolved nutrients, etc.) in the water from the two locations (surface and deep), which could represent a confounding factor in the interpretation of their results. Data are presented only for the deep water samples (see lines 350-351). The authors should present these data for both water types and discuss how it might have impacted the study.

Based on the mean and +/- data in Table 1, the release of TnBP looks to be significantly higher at 10 MPa than at 0.1 MPa, but in line 134 the authors describe this result as "slightly higher", implying that it is not significant, and in lines 235 to 237 they state "the results showed that hydrostatic pressure ... had no effect on the more hydrophilic substances." The authors should test this conclusion statistically.

Throughout the manuscript the authors state that "the addition of PVC ... decreased the number of bacteria" (e.g. see lines 194-195, 205, 207) but for the surface water the bacterial abundance in the PVC treatments remained consistent throughout the incubation, whereas the bacterial abundance in the control and PE increased. Preventing an increase is not the same as causing a decrease. The authors should be more careful in their description of this result. Based on this issue, I do not agree with the authors conclusion of a "toxic effect of plastic debris on natural bacterial assemblages as shown for PVC in Fig. 3" (see lines 323-324).

Is it possible that some of the leached organic additives could have been biodegraded by planktonic microbes in the biotic treatments, thus decreasing the concentrations of these compounds measured in the water?

Specific Comments

Line 24 I'm not sure what "standardized" means in this context, especially since the PE consisted of recycled low-density polyethylene (see line 354).

Line 106 Please clarify that "ng g⁻¹" refers to g of starting material.

Table 1 The table itself should indicate the units for the data values.

Table 1 Please explain the meaning of the +/- values, i.e. standard deviation or standard error.

Line 125 Change "are" to "were".

Figs 1, 2, 3, 4 The figure legend should indicate the number of replicates and whether the error bars represent standard deviation or standard error.

Line 153 I think the data value should be 738 (see Table 1) rather than 783.

Line 156 I'm not sure the meaning of "pure release". Does this refer to abiotic release? Please clarify.

Fig 3 It would have been nice to have data for the controls for the same time points as the treatments.

Fig 3 Why are there two data points for the controls, while the treatments are presented as mean values with error bars?

Fig 4 Why does the figure not include error bars?

Lines 370-378 It is unclear to me if all samples were incubated in the hyperbaric apparatus or if only the high pressure samples were incubated in this apparatus. Ideally all samples would have been incubated in this apparatus, with the pressure set differently for the high and low pressure incubations. If only the high pressure treatments were placed in the apparatus then the apparatus itself would be a confounding factor.

Line 360 Change "quality" to "source" or "location".

Line 402 The assay used to count cells would not be specific to only heterotrophic prokaryotes.

Reviewer #1 (Remarks to the Author):

Summary

The given paper claims on the leaching process of plastic additives. In this context preproduction pellets of two representative, pristine polymers, polyethylene (PE) and plasticized polyvinylchloride (PVC) were treated in a seawater environment for 30 days. Representative for surface and deep sea conditions the experiments were carried out at 0.1 MPa with surface and 10 MPa with deep sea water, respectively. Here the leaching processes were performed in the dark with seawater's natural prokaryote assemblage and abiotic seawater (HgII+ treatment). The four different conditions were compared focusing the influence of pressure and presence of prokaryotes on the leaching process. The leachates were analyzed for 23 additives, 7 phthalates (PAEs) and 9 organophosphate esters using GC/MS in SIM mode, and 7 bisphenols using LC/QTOF. As a result 3 phthalates and 2 organophosphates in PE leachates (in sum between 212 and 739 ng per g polymer), and BPS as only additive in PVC leachates (in sum 0.2 to 3.3 ng per g polymer) were identified and quantified. This led to the conclusion, that cumulative release of additives below ppm levels is orders of magnitudes lower than percent levels included during production. Representative for the PE leachate Tris(2-ethylhexyl) phosphate (TEHP) with high log Kow and dimethyl phthalate (DMP) with high and low log Kow were chosen to describe effects of pressure and prokaryotes. Additives were leached less at higher pressure conditions. This effect was particularly expressed for TEHP (=high Log Kow). For cumulative data the factor was around 2 suggesting a reduced additive release under increased water pressure. Additives were leached to a higher extent at biotic conditions at same pressure conditions, and more expressed under surface conditions in particular regarding TEHP. This draws the conclusion to an increased additive release at the presence of bacteria. Additionally 7 phthalate monoesters (MPAEs) metabolites (only two are specified, five are missing L 170/L393) were analyzed with LC/QTOF as well. Of those two were detected in the PVC leachate with highest concentration at biotic and surface conditions (146 ng per g polymer after 30 days) point to not targeted and therefore not analyzed PAEs. One MPAE in the PE biotic conditions leachates were detected as well (16 and 9 ng per g polymer). Their presence was led back to microbial biodegradation. The leachate of PVC led to decreased bacteria counts compared to PE leachate after 30 days and was led back to toxic effects of leached non-targeted OPEs or PAEs. All results were discussed, a model calculation of globally released additives based on the data and finally an outlook were given.

Remarks

The idea to perform additive leaching experiments under increased pressure (deep sea conditions) is interesting and so far not published. Same holds true for the parallel studies of biotic/abiotic conditions and its influence on additive leaching. Regarding the seafloor as a relevant sink for plastic litter and degradation products the analytical setup and achieved results are of general relevance. The paper is written in fluent and well readable English.

The observed results and trends are stimulating and worth to be followed but, nevertheless, very limited and not convincing on the presented level. Therefore the overall conclusions are not convincing and even partly misleading at the present stage.

The major point of critique regarding the study is the fact, that the authors did not characterize the pristine polymers used in the experiments with regard on their additive content in advance. From my point of view this is fundamental for this kind of study in particular when working with a model system.

The authors don't seem to have reviewed details about usual additive contents of specific polymers rather than on a very general view.

PE is a polymer with relatively low additive content (1-2%) and rarely plasticized. On the other hand soft PVC contains approximately 30% of plasticizers (33% (density 1.22), 44% (density 1.17), PVC-Handbook, Wilkes et al. 2005) and the stabilizer content (e.g. BPA or its substitute BPS) is << 1%. Most common plasticizers used for PVC are dialkyl phthalates (e.g. octyl, nonyl, decyl), technical mixtures of isomers, no easy to analyze but definitely to be included in such kind of study.

PVC is easily solvable in e.g. THF. It is no problem at all to identify and even quantify its containing additives after polymer precipitation using GC/MS (for the here relevant PAEs and related plasticizers) and of course in full scan mode. This is very easy step and obligate for an analytical question as it is asked by the authors. Working directly on a targeted (SIM)-level and analyzing the leachates only makes the analysis set up "blind" for other compounds and consequently might lead to misleading and partial results. Same holds true for PE, even if it is more difficult to get into solution, a severe leaching with appropriate solvent would have given valuable insights. Having this in mind it is highly doubtful that PE releases more additives like plasticizers than PVC (Table 1), and PVC leachate contains BPS, only. Consequently, Table 1 represents an incomplete picture of the analytical setup and conclusions like

L 122-123 "The cumulative release of additives from polymers was always below ppm levels..."
or

L 227-228 "PE released more organic additives than PVC, both in quantity and diversity..."
are not supported.

The given, relativizing argumentation in L228-230 "Although no measurable leaching of PAEs was observed from PVC, it is possible that non-targeted PAEs or OPEs may have leached from this polymer", even if it is true from the data sets point of view, doesn't change the fact that the given results reflect a distorted picture of the leachate. The additional argumentation (L 230 – 234) is not an excuse at all for the missing investigation and characterization of the additives representing at least 1/3 of the used soft PVC pellets by weight! Even though the authors know their lack of knowledge regarding the PVC, they underestimate its impact by far and make any further conclusion doubtful.

The presence and discussion of PAEs metabolites (L170ff) where those of PVC exceed those of PE by far underline the above critique even the argument of non-targeted PAEs is stressed again.

We first want to thank R1 for his or her in deep review of the manuscript, and we definitely believe his or her comments helped the manuscript being more informative and of interest. Please note that due to the health crisis, access to the instruments of the laboratory's chemistry platform has been slightly delayed.

We do not consider the ambition of getting a comprehensive overview of plastic leachates as an easy step. Comprehensive non targeted strategy implies the use of HRMS for GC, LC ESI positive, LC ESI negative analysis, and we think it is a matter of study by itself, given the still-limiting technical difficulties inherent to this strategy. Moreover, following the Rev#1 suggestions, we undertook to dissolve PVC and PE with appropriate protocols (dissolution of PVC in dichloromethane and recrystallisation by adding methanol, dissolution of PE in toluol at 150 °C and recrystallisation by adding methanol), but the extracts were too viscous to be injected in GC and LC-MS. Otherwise, these protocols might not allow being quantitative since i) additives partitioning between solvent phase and recrystallized polymer, and ii) thermodegradation for PE extracts may occur. Therefore, in recognition of the disqualifying weaknesses of the target approach, notably concerning the assumption that PE releases more additives than PVC, we reanalyzed all extracts in fullscan mode. For PE we did not find significant additional substances, but for PVC we found massive peaks of DiNP isomers at the end of the GC chromatogram. This finding, that we owe to R1, modifies significantly the conclusions mentioned above. These new informations are now included in the revised version and the manuscript was deeply modified as follows:

L27 "Here we show that in abiotic conditions increasing hydrostatic pressure inhibits the leaching of the heaviest organic additives such as tris(2-ethylhexyl) phosphate and diisononyl phthalate from polyethylene and polyvinylchloride materials, whereas deep-sea and surface marine prokaryotes promote the release of all targeted additives (phthalates, bisphenols, organophosphate esters)."

L77 "Our results show that (i) marine prokaryotes in surface seawater, and to a lesser extent in deep seawater, promote the release of PAEs, OPEs and BPs, and (ii) high hydrostatic pressure inhibits the leaching of the heaviest and more hydrophobic organic additives such as diisononyl phthalate (DiNP, PAE) and tris(2-ethylhexyl) phosphate (TEHP, OPE)."

L105 "Table 1. Organic additive release from polyvinylchloride (PVC) or polyethylene (PE) to seawater under contrasting conditions of hydrostatic pressure and bacteria content for 30 days. The values (n = 10 or 14) presented show the maximum additive release at the end of the exposure (i.e. plateau observed with first order kinetics modeling, ng g⁻¹ or µg g⁻¹ for DiNP). Each value is given ± standard deviation. Brackets contain the values for which a linear regression was applied (i.e. regression slope in ng of additive released per gram of polymer per day). The additives detected were dimethyl phthalate (DMP), diethyl phthalate (DEP), diethylhexyl phthalate (DEHP), diisononyl phthalate (DiNP), tri-n-butyl phosphate (TnBP), tris(2-ethylhexyl) phosphate (TEHP), bisphenol S (BPS). The two different hydrostatic pressures are equivalent to atmospheric pressure (0.1 MPa) and 1000 m depth (10 MPa). Deep seawater at 0.1 MPa under biotic conditions was not investigated because it was not relevant for natural deep prokaryote development"

	Treatment	A	B	C	D	E	
	Seawater source	Surface	Surface	Deep	Deep	Deep	Additive
	Pressure (MPa)	0.1	0.1	0.1	10	10	molecular
Polymer	Additive \ Bacteria	biotic	abiotic	abiotic	abiotic	biotic	weight (Da)
PE	DMP (ng g ⁻¹)	38.0 ± 6.2	13.8 ± 1.9	14.7 ± 3.3	12.9 ± 0.2	24.8 ± 5.9	194
PE	DEP (ng g ⁻¹)	22.8 ± 1.5	8.0 ± 0.9	7.4 ± 0.7	5.9 ± 0.6	13.9 ± 2.7	222
PE	TnBP (ng g ⁻¹)	421 ± 24	183 ± 12	149 ± 10	209 ± 14	302 ± 19	266
PE	DEHP (ng g ⁻¹)	222 ± 25	64.3 ± 27.0	33.6 ± 5.9	8.9 ± 3.7	31.1 ± 0.2	390
PE	TEHP (ng g ⁻¹)	34.7 ± 4.2	8.3 ± 0.7	7.8 ± 0.5	< 1.6	2.4 ± 1.0	434
PE	Cumulative (ng g ⁻¹)	738.5	277.4	212.5	236.7	374.2	
PVC	BPS (ng g ⁻¹)	[0.2 ± 0.1]	2.5 ± 0.4	2.7 ± 0.3	3.3 ± 0.4	[0.2 ± 0.1]	250
PVC	DiNP (µg g ⁻¹)	88.2 ± 23.1	17.6 ± 2.4	18.2 ± 3.7	4.1 ± 0.6	13.7 ± 4.2	418

L121 "In addition, DiNP and BPS were detected in leachates from the PVC samples".

L126 "The cumulative additive release from PE was always below ppm levels, irrespective of the biotic conditions and hydrostatic pressure applied. BPS release from PVC was below ppm levels for all conditions, whereas DiNP release from PVC was measured in the range 4-88 µg g⁻¹, which is up to two orders of magnitude higher than the cumulative release of all additives observed for PE. In both cases however, the additive release was far below the typical level initially included in the polymers during manufacture (1-10% for PE vs. up to 70% for PVC)¹⁶."

L139 "Under abiotic conditions, PE and PVC exposed to atmospheric (0.1 MPa) and high pressure (10 MPa) within HPBs exhibited additive leaching to seawater ranged from several ng to 4.1 ± 0.6 µg for individual compound per gram of polymer over the 30 day experiment (Table 1, treatments C and D). The 95% confidence intervals overlapped for the low molecular

weight additives such as BPS and the two PAEs (DMP and DEP). In contrast, the 95% confidence intervals did not overlap for higher molecular weight additives such as the OPE TEHP and the PAEs DEHP and DiNP (Supplementary Table 1, Fig. 1), meaning their leaching decreased significantly with increasing hydrostatic pressure."

L162 "Irrespective of the hydrostatic pressure applied (0.1 or 10 MPa) and the nature of the chemical considered, the presence of prokaryotes adapted to deep-sea conditions, in their original chemical environment significantly increased the amount of additives released from plastic pellets into the dissolved phase (separated 95% confidence intervals; Supplementary Table 1, Fig. 2)."

L188 "The release of MMP from PVC could be attributed to biodegradation of DiNP since it was demonstrated that PAEs with longer side chains can be converted to shorter chains by β -oxidation and eventually transesterification^{37,38}."

L221 "Among them, the amount of DiNP (13.7 to 88.2 $\mu\text{g g}^{-1}$ of PVC after 30 days of exposure, i.e. 527 to 3392 $\mu\text{g L}^{-1}$ under the operating conditions employed; Table 1) and BPS (6 ng g^{-1} of PVC after 30 days of exposure, i.e. 230 ng L^{-1} under the operating conditions employed; Table 1) quantified in the dissolved fraction could be responsible for all or part of the observed limitation in prokaryotes development compared to the control and PE treatments (Fig. 3). Interestingly, under surface and biotic conditions illustrated in Fig. 4, the release of DiNP (88.2 $\mu\text{g per gram of PVC}$, i.e. 65.8 $\mu\text{g C per gram of PVC}$; Table 1) accounted for 18% of the measured total DOC release (30.7 $\mu\text{mol g}^{-1}$, i.e. 368 $\mu\text{g C per gram of PVC}$) under the surface and biotic conditions illustrated in Fig. 4."

L244 "The study did not aim to provide a comprehensive characterization of the mixture of organic additives present in the polymers studied. However, based on the list of 25 target additives, it was evident that (i) PVC released two orders of magnitude more additives than PE, (ii) DiNP was the main compounds measured in PVC and that OPEs were the predominant chemical family leached from PE, and (iii) PE released a wider diversity of additives."

L277 "The cumulative release of target plastic additives in surface and deep seawaters was found to range from 212 to 738 ng g^{-1} for PE and from 4.1 to 88.2 $\mu\text{g g}^{-1}$ for PVC"

L288 "These results, which are consistent with the available but sparse literature on the topic²⁷, are 2-3 orders of magnitude lower than the 6% additive content (of which 50% is organic chemicals⁴²) mentioned for the global plastic manufacturing data¹."

L413 "DiNP was also added to the target list and analyzed in the first fraction by GC-MS, after first being identified in the PVC leachates by analyses performed in fullscan mode."

L 257ff General calculations on overall additive releases into the ocean founded on the given database seem to be very vague and doubtful in particular regarding the PVC share. Paluselli et al. 2019 analyzed only a very restricted number of PAEs and not the leachate in general; their study has the same weak point and should be handled with caution as well regarding some of the given generalized results, disregarding some very interesting and fundamental other aspects presented there. Literature data cited by the authors as well reported relevant higher additive leaching for PVC (upper $\mu\text{g-range per g polymer}$, Suhrhoff and Scholz-Böttcher, 2016) in model systems. The argument that non targeted additives are missing should consequently exclude respective calculations.

Although DiNP was included in the list, the text was modified as follows:

L281 "Considering the estimated levels of plastic discharges to the global ocean proposed by Jambeck et al.⁴, together with the polymer manufacturer's distribution mentioned by Geyer et al.¹, it is possible to propose tentative additive chemical release estimates of 2.3-132 tons per year from PVC, and 0.4-3.4 tons per year from PE"

L 323ff The conversion of the observed cell number decrease into a toxic effect of plastic debris on natural bacterial assemblages is a fair hypothesis but should not be dramatically overdrawn as written in L327-329 "As a result, the release of a cocktail of substances, including additives, oligomers, and polymer-based nanoparticles might be responsible for an overall decrease in bacteria content over time due to toxic effects". Is there any indication, that there is such an effect (as implied by the sentence already) and is it realistic that (the very low) plastic abundance in sediment release a toxic leachate level by the state of knowledge so far? Here, a more hypothetical and scientific grounded argumentation would be preferred. At the presented level the data do not allow that high level of generalization regarding the possible impact of additive leaching in deep sea environment (L 278 ff).

Yes, evidence here is not sufficient to conclude a toxic effect. The manuscript has been modified accordingly:

L221 "Among them, the amount of DiNP (13.7 to 88.2 $\mu\text{g g}^{-1}$ of PVC after 30 days of exposure, i.e. 527 to 3392 $\mu\text{g L}^{-1}$ under the operating conditions employed; Table 1) and BPS (6 ng g^{-1} of PVC after 30 days of exposure, i.e. 230 ng L^{-1} under the operating conditions employed; Table 1) quantified in the dissolved fraction could be responsible for all or part of the observed limitation in prokaryotes development compared to the control and PE treatments (Fig. 3)."

L339 "The role of prokaryotes in plastic degradation could also be influenced by their growth limitation induced by plastic debris, both for surface and deep communities, as shown for PVC in Fig. 3."

L343 "As a result, the release of a cocktail of substances, including additives, oligomers and polymer-based nanoparticles might be responsible for an overall growth limitation of prokaryotes over time, as highlighted otherwise.^{53"}

The same holds true regarding the conclusion that "biodegradation is the main driver for plastic degradation and additive release in deep-sea environments" (L302 ff). Regarding the presented data no clear relation to a polymer degradation process is documented. The discussion is suggested to be performed more result-related, more hypothetical and less general.

The text has been modified:

L81 "This study provides the first empirical evidence of a more efficient additive release at the ocean surface"

L323 "As shown by the current study, the action of prokaryotes appears to be the main driver for additive release in deep-sea environments."

The analytical equipment used by the authors would allow a consistent and comprehensive experimental data set. A repetition of the experiment with a profound prior analysis of the additive content of the "representatively" used polymers and a subsequent related analysis of the leachates first in full scan MS mode and successively on an adapted set of targeted additives will considerably improve the quality of the results. Subsequently derived conclusions are than supposed to be much more profound and convincing for superordinate conclusions. In the current state I would recommend a rejection of the paper, according to the argumentation above. If the leachates are still available they should be reanalyzed targeting those additives identified in the PVC/PE used for the experiments in an obligate first step. Analysis of leachates should be performed in full scan and SIM mode as well with regard to further degradation products (e.g. related MPAs). An accordingly revised manuscript should give a much more convincing picture of the experimental setup and derived conclusions. In this case a resubmission should be offered.

We believe we have addressed most important concerns raised by the reviewer. We appreciated the help of the reviewer. Several organic leachate compounds could be identified and included in the revised version of the MS. Although our study probably does not include all of the additive compounds and metabolites, we believe that we provide innovative information on the degradation of plastics and leaching of organic additives in the deep ocean.

Reviewer #2 (Remarks to the Author):

Recommendation: Accept with minor revisions

General Comments

This paper presents results of lab experiments that measured the amounts of phthalates, organophosphates and bisphenols leached from polyethylene and PVC at different pressures and in the presence and absence of prokaryotes. The results are interesting in that they provide some clues about how the persistence of plastic materials is affected by sinking in the ocean.

The paper is well written and clearly presented. I support its publication after the authors consider my specific comments and suggestions below.

We thank R2 for his or her clear summary of the manuscript.

Specific Comment

Line 41: Another study in addition to (13) that shows polymer chain scission is a source of DOC is this recent paper by Colin Ward & Chris Reddy (<https://pubs.acs.org/doi/abs/10.1021/acs.estlett.9b00532>)

L45: reference added

Line 46: "Most additives that have been detected in the ocean are endocrine disruptors... food web transfer toward higher trophic level organisms". Looking at the references for this sentence, I think the authors are conflating different issues and lumping them together in a way that will mislead readers and probably overstates the (known) problem. The statement that "most are EDCs" seems to be referring to phthalate esters and bisphenol-A. But, phthalates and bisphenol-A are not bioaccumulative, and although they are transferred to higher trophic levels they undergo "trophic dilution" and are thus not considered a high priority due to B (see for example <https://pubs.acs.org/doi/full/10.1021/es034745r> and <https://doi.org/10.1016/j.chemosphere.2015.08.059>).

L53. We agree the sentence was misleading, and thank R2 for the ref. However, the references initially included in the manuscript show additive transfer to living organisms, although at the first levels of the marine food web (i.e. zooplankton). Therefore, we removed "toward higher trophic level organisms".

Line 78: "This study provides the first empirical evidence of a more efficient plastic debris degradation and subsequent additive release at the surface than in deep-sea..." I think the wording here is a little too loose. I think it is plausible that this is the first study showing more efficient additive release at the surface. But I am quite confident there are studies out there that provide empirical evidence that degradation of plastic debris is more efficient at the surface (due to UV light exposure if nothing else, as the authors themselves write on lines 250-252!).

We agree the link between plastic degradation and additive release is not straightforward and "plastic debris degradation and subsequent" was removed (L82).

Line 81: The Results section is very nicely organized and presented!

Thanks for positive feedback.

Figure 2: As a reader, I want to make visual comparisons between the abiotic and biotic treatments, but also between the two pressures. So the data in the upper and lower panels for TEHP and DMP should be presented on the same vertical scale. This will not be a problem at all for DMP. For TEHP the data at 10 MPa will be close to the bottom of the plot, but I think that is OK since there is no trend in the data anyway...

L178/Fig.2: thanks for helpful remark. The vertical scales have been harmonized.

Figure 3: As above, I would like to see these two panels presented with the same vertical scale so I can instantly visually compare the surface seawater data with the deep seawater data.

L212: The vertical scales have been harmonized.

Line 297–307: This paragraph is a nice synthesis of when and where degradation of plastic polymers can happen in aquatic systems.

Thanks for positive feedback.

Line 315: "Monitoring..." This is an interesting idea, but my first reaction is that it will not work because there are so many sources of phthalates to the environment. I somehow doubt that leaching from plastic in-situ in the environment is a dominant source of phthalates in water such that the concentrations (or even the ratios of concentrations of monoesters to diesters) could be used to map plastic pollution. Maybe the authors could do some back-of-the envelope calculations based on their experiments and compare them to phthalate concentrations reported in the environment to check if their suggestion is reasonable.

L346: This sentence was not related anyway with the data presented here. We completely consider this comment, and preferred removing the sentence rather than providing highly putative information.

Line 511: There is an amusing typo in the title of Ref. 27 (FI Oating).

L555, ref 28: it has been corrected

Reviewer #3 (Remarks to the Author):

General Comments

This manuscript describes a study designed to compare the leaching of organic additives from microplastic to marine water at the ocean surface as compared to the deep ocean. Two types of plastic beads (polyethylene and polyvinylchloride) were incubated with sea water collected from the ocean surface (0.5 m) or at 1000 m depth. Samples were incubated under two pressure scenarios, 0.1 and 10MPa, for 30 days and the release of 23 organic additives was quantified. The manuscript is well written and clear and the results are presented clearly and effectively. The study produced some interesting results, including that the leaching of heavy molecular weight organic additives was lower under high pressure, and that the presence of microbes increased the release of organic additives.

My main concerns about this manuscript relate to experimental design, replication, and application of statistics. The study is attempting to address several independent variables (pressure, seawater source, presence vs. absence of live bacteria) but the design is not fully crossed, e.g. there were no incubations of surface water at high pressure and no biotic incubations of deep water at low pressure. A fully crossed design would have enabled a more rigorous statistical analysis of their data, e.g. using analysis of variance, which would have assessed individual effects of independent variables and interactions. As it stands the statistical analysis of their data is limited to a few haphazardly applied t-tests which are only mentioned parenthetically in the Results section. Moreover, the degree of replication included in the study is unclear. Table 1 states that their $n = 10$ or 14 , but in line 364 it states "each sample was deployed in duplicate". If the degree of replication was duplicates for each treatment, this is not adequate for a rigorous statistical analysis. This problem is exacerbated by the fact that there is no mention of statistics in the Methods section. The authors need to explain more clearly how their data were statistically analyzed and to address the limitations of their study design.

Yes, we agree with the reviewer on the fact that some sample treatment is lacking to achieve a fully crossed design of the experiment, and then perform ANOVA and discuss the interactions between parameters. We acknowledge that the limited amount of strictly identical experimental replicates ($n=2$) hampered a more robust statistical analysis. Nevertheless, we adjusted a first order kinetic to the overall kinetic, including 5 to 7 time points in duplicate (10 to 14 datapoints). Table 1 (L105) is the result of that modeling. Each data presented in this table was thus fitted on 10 or 14 datapoints. We think conclusions in the manuscript are in line with the design applied, and that no over interpretation of the data was done.

As data presented in Table 1 are the result of a data modeling (first order kinetic adjustment based on 10 or 14 independent sample points, we have clarified that in the revised version of the MS, cf. below), a non-parametric test was run (Mann-Whitney), which did not show any significant difference between the treatments applied. Even if differences were not statistically significant (most probably due to the limited amount of data and power of the non parametric test) some trends are quite clear. We therefore decided to compare the 95% confidence intervals in order to further explore significant differences between treatments. As this method is very straightforward, we did not add a specific method section for statistics. Results of this analysis are presented in SI (Supplementary Table 1) of the revised MS.

The revised manuscript was modified as follows:

L139: "Under abiotic conditions, PE and PVC exposed to atmospheric (0.1 MPa) and high pressure (10 MPa) within HPBs exhibited additive leaching to seawater ranged from several ng to $4.1 \pm 0.6 \mu\text{g}$ for individual compound per gram of polymer over the 30 day experiment (Table 1, treatments C and D). The 95% confidence intervals overlapped for the low molecular weight additives such as BPS and the two PAEs (DMP and DEP). In contrast, the 95% confidence intervals did not overlap for higher molecular weight additives such as the OPE TEHP and the PAEs DEHP and DiNP (Supplementary Table 1, Fig. 1), meaning their leaching decreased significantly with increasing hydrostatic pressure. One exception is TnBP, whose release seemed to increase with increasing pressure."

L162: "Irrespective of the hydrostatic pressure applied (0.1 or 10 MPa) and the nature of the chemical considered, the presence of prokaryotes adapted to deep-sea conditions, in their original chemical environment significantly increased the amount of additives released from plastic pellets into the dissolved phase (separated 95% confidence intervals; Supplementary Table 1, Fig. 2)".

As regards replication, we agree the manuscript was unclear for the reader. Each sample was deployed in duplicate, but kinetic parameters were modeled with 5 or 7 time points in duplicates. This was clarified along the revised version of the manuscript:

L107: "The values show the maximum additive release at the end of the exposure, when a plateau is observed with first-order kinetic modeling based on 10 or 14 independent data (ng g^{-1} for all chemicals except DiNP, which is expressed in $\mu\text{g g}^{-1}$)."

L380: "All experiments were performed in the dark at $13 \text{ }^\circ\text{C}$ in a temperature-controlled laboratory and each condition was deployed in duplicate. The kinetic parameters shown in Table 1 were therefore determined on 10 or 14 datapoints (corresponding to 5 or 7 time points in duplicates)."

The study assumes that the prokaryotic communities from 0.1 m and 1000 m are different in taxonomic composition. This is almost certainly true, but it would have been nice to have seen this documented, for example via 16S amplicon sequencing.

Yes, we agree. Sampling at 1000 m was performed in June 2018 in the NW Mediterranean Sea when water stratification occurs. This implies prokaryotic differences between surface and deep communities (Tamburini et al. 2013. Prokaryotic responses to hydrostatic pressure in the ocean – a review. *Environmental Microbiology* 15(5), 1262-1274). Stratification is supported by the CTD data. The text was modified accordingly in the revised MS:

L160: "Seawater sampling was carried out in late spring, at the time of water mass stratification (see Supplementary Figure 2), which involves different prokaryotic assemblages between deep and surface seawater.³³".

L367: "Deep seawater and surface seawater had the following characteristics, respectively: salinity = 38.5 and 37.9, DOC = 1.3 and 7.2 mg C L⁻¹".

The study does not present, account for, or discuss the possible effects of physical or chemical differences (e.g. pH, osmotic concentration, dissolved nutrients, etc.) in the water from the two locations (surface and deep), which could represent a confounding factor in the interpretation of their results. Data are presented only for the deep water samples (see lines 350-351). The authors should present these data for both water types and discuss how it might have impacted the study.

The data are now indicated for both types of seawater:

Yes we fully agree with R3 that the impact of water chemistry was not discussed. From an abiotic point of view, the release of additives is not impacted by the water chemistry (cf. treatment B vs. C, Fig 1, L105). Nevertheless, the differences in dissolved nutrients concentration between deep and surface waters may have affected the development of bacteria. This is now discussed in the revised version of the MS:

L131: "Water chemistry under abiotic conditions (treatment B vs. C; Fig. 1) was not found to play a significant role in influencing additive release."

L157: "In addition to the abiotic pressure studies, two complementary experiments were performed with surface and deep seawater where the natural prokaryotic assemblage together with the in situ water chemistry were maintained and investigated as a potential driver for organic additive release from PE and PVC (Table 1, Figs. 2, 3). Seawater sampling was carried out in late spring, at the time of water mass stratification (see Supplementary Figure 2), which involves different prokaryotic assemblages between deep and surface seawater.³³ Irrespective of the hydrostatic pressure applied (0.1 or 10 MPa) and the nature of the chemical considered, the presence of prokaryotes adapted to deep-sea conditions, in their original chemical environment significantly increased the amount of additives released from plastic pellets into the dissolved phase (separated 95% confidence intervals; Supplementary Table 1, Fig. 2)."

L264: "The presence of heterotrophic prokaryotes in their original chemical environment was also found to promote additive release".

L362: "Deep-sea seawater and surface seawater had the following characteristics, respectively: salinity = 38.5 and 37.9, DOC = 1.3 and 7.2 mg C L⁻¹".

L367: "Deep seawater and surface seawater had the following characteristics, respectively: salinity = 38.5 and 37.9, DOC = 1.3 and 7.2 mg C L⁻¹".

Based on the mean and +/- data in Table 1, the release of TnBP looks to be significantly higher at 10 MPa than at 0.1 MPa, but in line 134 the authors describe this result as "slightly higher", implying that it is not significant, and in lines 235 to 237 they state "the results showed that hydrostatic pressure ... had no effect on the more hydrophilic substances." The authors should test this conclusion statistically.

Yes the difference is significant, and we do not have any explanation for this. This is now indicated in the revised MS:

L147: "Although significant (non-overlapping 95% confidence intervals), this difference in behaviour compared to the other chemicals studied is difficult to explain."

Throughout the manuscript the authors state that "the addition of PVC ... decreased the number of bacteria" (e.g. see lines 194-195, 205, 207) but for the surface water the bacterial abundance in the PVC treatments remained consistent throughout the incubation, whereas the bacterial abundance in the control and PE increased. Preventing an increase is not the same as causing a decrease. The authors should be more careful in their description of this result. Based on this issue, I do not agree with the authors conclusion of a "toxic effect of plastic debris on natural bacterial assemblages as shown for PVC in Fig. 3" (see lines 323-324).

Yes we agree, evidences here were not sufficient to conclude a toxic effect. The manuscript has been modified accordingly:

L221 "Among them, the amount of DiNP (13.7 to 88.2 µg g⁻¹ of PVC after 30 days of exposure, i.e. 527 to 3392 µg L⁻¹ under the operating conditions employed; Table 1) and BPS (6 ng g⁻¹ of PVC after 30 days of exposure, i.e. 230 ng L⁻¹ under the operating conditions employed; Table 1) quantified in the dissolved fraction could be responsible for all or part of the observed limitation in prokaryotes development compared to the control and PE treatments (Fig. 3)."

L339 "The role of prokaryotes in plastic degradation could also be influenced by their growth limitation induced by plastic debris, both for surface and deep communities, as shown for PVC in Fig. 3."

L343 "As a result, the release of a cocktail of substances, including additives, oligomers and polymer-based nanoparticles might be responsible for an overall growth limitation of prokaryotes over time, as highlighted otherwise.⁵³".

Is it possible that some of the leached organic additives could have been biodegraded by planktonic microbes in the biotic treatments, thus decreasing the concentrations of these compounds measured in the water?

This discussion is already included in L167-176, and states that our results may underestimate the additive release by a maximum of 50%.

Specific Comments

Line 24 I'm not sure what "standardized" means in this context, especially since the PE consisted of recycled low-density polyethylene (see line 354).

Standardized means the main PE characteristics are described by the supplier. It has been removed to avoid confusion in the revised MS. Details on each polymer are otherwise given in the material section L371-374 of the revised MS.

Line 106 Please clarify that "ng g⁻¹" refers to g of starting material.

L105/Table 1: this has been specified.

Table 1 The table itself should indicate the units for the data values.

L118/Table 1: Done.

Table 1 Please explain the meaning of the +/- values, i.e. standard deviation or standard error.

L105/Table 1: this has been specified.

Line 125 Change "are" to "were".

L134: Done.

Figs 1, 2, 3, 4 The figure legend should indicate the number of replicates and whether the error bars represent standard deviation or standard error.

Done.

Line 153 I think the data value should be 738 (see Table 1) rather than 783.

L157: Yes, absolutely. This sentence however is no more in the corrected version.

Line 156 I'm not sure the meaning of "pure release". Does this refer to abiotic release? Please clarify.

L168: "pure release" has been changed to "net flux". This means the apparent quantity of additive measured resulted from the net flux from plastic to water, minus the degradation that may occur in the water phase.

Fig 3 It would have been nice to have data for the controls for the same time points as the treatments.

Unfortunately, we do not have these samples or data. We agree it would have been nice, although the additive comparison is made on the plateau observed at the end of the experiment.

Fig 3 Why are there two data points for the controls, while the treatments are presented as mean values with error bars?

L213/fig3: controls are now presented as mean value with associated standard deviation error bars in the revised MS.

Fig 4 Why does the figure not include error bars?

L234/fig4: Error bars were omitted; they have been added in the revised MS.

Lines 370-378 It is unclear to me if all samples were incubated in the hyperbaric apparatus or if only the high pressure samples were incubated in this apparatus. Ideally all samples would have been incubated in this apparatus, with the pressure set differently for the high and low pressure incubations. If only the high pressure treatments were placed in the apparatus then the apparatus itself would be a confounding factor.

L384: every sample was incubated in a 130-mL glass bottle (high and low pressure). For high pressure samples, these bottles were immersed in high pressure bottles. Therefore, all sample containers were the same (130-mL glass bottle). It is now clarified in L389 in the revised MS: "Samples set at atmospheric pressure (0.1 MPa) were not incubated in HPBs."

Line 360 Change "quality" to "source" or "location".

L378: Done.

Line 402 The assay used to count cells would not be specific to only heterotrophic prokaryotes.

Cytometry indeed allows the counting of both auto- and heterotrophic cells. Our protocol however allowed identifying and removing autotrophic cells by analyzing the red fluorescence induced by chlorophyll *a* (Marie, D., F. Partensky and D. Vault. 1996. – Application of the novel DNA dyes YOYO-1, YOPRO-1 and Picogreen for flow cytometric analysis of marine prokaryotes. Appl. Environ. Microbiol., 62: 1649-1655).

Reviewers' comments:

Reviewer #1 (Remarks to the Author):

As already written in the former review (expected to be on hand and therefore not repeated), the idea to perform additive leaching experiments under increased pressure (deep-sea) conditions is innovative. Combined with the studies of biotic/abiotic conditions and their influence on additive leaching the experimental design is original. The revised data set makes the given results, trends and interpretation much more convincing. It would have been perfect, in particular, regarding any statements concerning the mass balance/leaching efficiency, to have a consistent additive data set for the pristine polymers as well. Nevertheless, this lack does not weaken the general and interesting observations made by the authors.

The ocean, its water-column, and in particular the (deep) seafloor is a relevant sink for plastic litter and its degradation products. The presented insights into the leaching behavior of plastic additives decreased at higher water pressure but increased under biotic conditions are of general relevance. The authors have overall improved their manuscript. They have taken almost all of my concerns of the former review into consideration and incorporate them into their manuscript. From my point of view, it is definitely recommendable to be published in nature communications, now.

Minor but desirable revision concerning table 1:

1. Please add the cumulative additive amount for PVC as well, and here preferably in ng g^{-1} (same dimensions as chosen for PE) to highlight the difference in dimension of leached additives visibly.

2. Please exchange the order of TnBP and DEHP, to group phthalates and phosphates, respectively, since the abbreviations are not ideal. A clarifying, additional option would be to insert "sum of phthalates" and "sum of phosphates" as well to underline their respective proportion.

Reviewer #2 (Remarks to the Author):

I was Reviewer #2 of the original manuscript. My opinion after reviewing all three reviewer's comments and the authors responses is that this seems like a very good example of the peer review process helping the authors to improve their manuscript! I support publication of the revised manuscript.

Reviewer #3 (Remarks to the Author):

I served as reviewer #3 on the prior version of this manuscript. The authors have responded to my critiques of the prior version, but their responses have not demonstrated that their data were analyzed using appropriate and rigorous statistical analyses. In their response the authors state "We acknowledge that the limited amount of strictly identical experimental replicates ($n=2$) hampered a more robust statistical analysis." Having only two experimental replicates for a controlled laboratory-based study conducted in bottles is inadequate, and as the authors admit, does not permit rigorous statistical analysis. In my view this is a fatal flaw in this study. The authors explained their data analysis as follows: "we adjusted a first order kinetic to the overall kinetic, including 5 to 7 time points in duplicate (10 to 14 datapoints). Table 1 (L105) is the result of that modeling. Each data presented in this table was thus fitted on 10 or 14 datapoints." The authors calculated a maximum additive release within each of their bottles based on 5 to 7 time points per bottle. In the legend of Table 1 they describe these time points as "10 or 14 independent data", but they are clearly not independent data, as they are 5-7 repeated samplings from the same bottle. Furthermore, what they end up with is one value for maximum additive release for each bottle, so in terms of statistical analysis, their degree of replication remains 2, which again is not enough replication for rigorous statistical analysis, e.g. via

an ANOVA.

In their response the authors go on to explain that "As data presented in Table 1 are the result of a data modeling (first order kinetic adjustment based on 10 or 14 independent sample points, we have clarified that in the revised version of the MS, cf. below), a non-parametric test was run (Mann-Whitney), which did not show any significant difference between the treatments applied." So statistical analysis of their data did not show any significant differences between the treatments. That to me is the key finding. No differences. However, the authors state "Even if differences were not statistically significant (most probably due to the limited amount of data and power of the non parametric test) some trends are quite clear. We therefore decided to compare the 95% confidence intervals in order to further explore significant differences between treatments." This is completely inappropriate. Since the statistics did not show a significant result they decided on significance just by looking at the data. This does not represent a rigorous and unbiased analysis of the data.

The authors conclude that "We think conclusions in the manuscript are in line with the design applied, and that no over interpretation of the data was done." I strongly disagree and suggest that none of the conclusions of their study are supported by the results, since statistical analysis, as they state "did not show any significant differences between the treatments."

Manuscript NCOMMS-19-39508

N.B.: all authors inputs and answers are noted in blue in this document and the associated revised manuscript.

Reviewer #1 (Remarks to the Author):

As already written in the former review (expected to be on hand and therefore not repeated), the idea to perform additive leaching experiments under increased pressure (deep-sea) conditions is innovative. Combined with the studies of biotic/abiotic conditions and their influence on additive leaching the experimental design is original. The revised data set makes the given results, trends and interpretation much more convincing. It would have been perfect, in particular, regarding any statements concerning the mass balance/leaching efficiency, to have a consistent additive data set for the pristine polymers as well. Nevertheless, this lack does not weaken the general and interesting observations made by the authors.

The ocean, its water-column, and in particular the (deep) seafloor is a relevant sink for plastic litter and its degradation products. The presented insights into the leaching behavior of plastic additives decreased at higher water pressure but increased under biotic conditions are of general relevance.

The authors have overall improved their manuscript. They have taken almost all of my concerns of the former review into consideration and incorporate them into their manuscript. From my point of view, it is definitely recommendable to be published in nature communications, now.

The authors thank R1 for their help and sound review.

Minor but desirable revision concerning table 1:

1. Please add the cumulative additive amount for PVC as well, and here preferably in ng g^{-1} (same dimensions as chosen for PE) to highlight the difference in dimension of leached additives visibly.
2. Please exchange the order of TnBP and DEHP, to group phthalates and phosphates, respectively, since the abbreviations are not ideal. A clarifying, additional option would be to insert "sum of phthalates" and "sum of phosphates" as well to underline their respective proportion.

- Page 5, Line 100: Table 1 has been modified accordingly.

Reviewer #2 (Remarks to the Author):

I was Reviewer #2 of the original manuscript. My opinion after reviewing all three reviewer's comments and the authors responses is that this seems like a very good example of the peer review process helping the authors to improve their manuscript! I support publication of the revised manuscript.

The authors are grateful to R2 for their positive feedback.

Reviewer #3 (Remarks to the Author):

I served as reviewer #3 on the prior version of this manuscript. The authors have responded to my critiques of the prior version, but their responses have not demonstrated that their data were analyzed using appropriate and rigorous statistical analyses. In their response the authors state "We acknowledge that the limited amount of strictly identical experimental replicates ($n=2$) hampered a more robust statistical analysis." Having only two experimental replicates for a controlled laboratory-based study conducted in bottles is inadequate, and as the authors admit, does not permit rigorous statistical analysis. In my view this is a fatal flaw in this study. The authors explained their data analysis as follows: "we adjusted a first order kinetic to the overall kinetic, including 5 to 7 time points in duplicate (10 to 14 datapoints). Table 1 (L105) is the result of that modeling. Each data presented in this table was thus fitted on 10 or 14 datapoints." The authors calculated a maximum additive release within each of their bottles based on 5 to 7 time points per bottle. In the legend of Table 1 they describe these time points as "10 or 14 independent data", but they are clearly not independent data, as they are 5-7 repeated samplings from the same bottle. Furthermore, what they end up with is one value for maximum additive release for each bottle, so in terms of statistical analysis, their degree of replication remains 2, which again is not enough replication for rigorous statistical analysis, e.g. via an ANOVA.

In their response the authors go on to explain that "As data presented in Table 1 are the result of a data modeling (first order kinetic adjustment based on 10 or 14 independent sample points, we have clarified that in the revised version of the MS, cf. below), a non-parametric test was run (Mann-Whitney), which did not show any significant difference between the treatments applied." So statistical analysis of their data did not show any significant differences between the treatments. That to me is the key finding. No differences. However, the authors state "Even if differences were not statistically significant (most probably due to the limited amount of data and power of the non-parametric test) some trends are quite clear. We therefore decided to compare the 95% confidence intervals in order to further explore significant differences between treatments." This is completely inappropriate. Since the statistics did not show a significant result they decided on significance just by looking at the data. This

does not represent a rigorous and unbiased analysis of the data.

The authors conclude that "We think conclusions in the manuscript are in line with the design applied, and that no over interpretation of the data was done." I strongly disagree and suggest that none of the conclusions of their study are supported by the results, since statistical analysis, as they state "did not show any significant differences between the treatments."

We thank R3 for providing relevant judgment about how statistics were performed and presented in the previous version of the manuscript and response to referees. We however believe that several misunderstandings led R3 to their decision. We believe that these misunderstandings might have originated from i) incomplete depiction of the experiment itself, and ii) incomplete description of the chosen statistical strategy.

- ⇒ *"Having only two experimental replicates for a controlled laboratory-based study conducted in bottles in inadequate, and as the authors admit, does not permit rigorous statistical analysis. In my view this is a fatal flaw in this study."*
- ⇒ *"but they are clearly not independent data, as they are 5-7 repeated samplings from the same bottle. Furthermore, what they end up with is one value for maximum additive release for each bottle, so in terms of statistical analysis, their degree of replication remains 2"*

All datapoints are truly independent samples (not subsampling at different times in 2 bottles, but a series of individual bottles). Replication at each time is 2, but replication over the kinetic is 10 to 14. The experiment was built in this way to fit the hyperbaric equipment (no high volume hyperbaric bottles available), and to avoid depressurization/pressurization of the samples at each sampling time, as this would have affected the microbiome within the experiment. All samples consisted of 130 mL bottles, sacrificed (in duplicate) after 0, 5, 10, 15, 20, 25 or 30 days of exposure. The entire 130 mL volume was utilized for chemical and biological analyses. The text has been clarified in the latest version accordingly:

- Page 5, Line 87: At each sampling time, two independent 130-mL bottle samples were sacrificed for chemical and biological analyses.
 - Page 6, Line 104, table 1 caption: individual 130 mL samples in duplicate sacrificed at each sampling time for each treatment,
 - Page 18, Line 380: At each sampling time, two 130 mL bottles per treatment were sacrificed, and the total volume was utilized for analyses in the following manner: 2 mL used for microbiological analysis, with 10 mL used for DOC analysis, 48 mL used for N and P analysis, and the remaining 70 mL kept at -20 °C until analyzed for the plastic additive content.
- ⇒ *"So statistical analysis of their data did not show any significant differences between the treatments. That to me is the key finding. No differences."*

Confidence intervals clearly show statistical differences between some of the treatments (see next paragraph). We previously applied a non-parametric equivalent of the ANOVA (Mann-Whitney and then Kruskal-Wallis for comparing 2 and 5 samples, respectively, not shown in the previous manuscript version, but in the previous response to referees) on the whole kinetic dataset, from t_0 to t_{final} , which was the only way to properly perform such a test on the data, as recommended by R3. These tests did not show any differences between treatments, with only few exceptions. These tests however did not compare the "maximum additive releases" as shown in Table 1, but the overall kinetics, which obviously, are very similar during the kinetic phase at the beginning of the experiment (i.e. before half time to equilibrium), whereas they tend to be distinct when approaching equilibrium.

- ⇒ *"This is completely inappropriate. Since the statistics did not show a significant result they decided on significance just by looking at the data"*

In order to reinforce the statistical part of the article, we decided to call on the skills of Dr David Nerini from MIO Aix-Marseille University (now listed as an author, expert in applied statistics for marine sciences) to rebuild our statistics strategy. The subsequent statistical analyses showed that a 95% confidence interval comparison between treatments along the kinetics duration is very appropriate. The statistical approach is now detailed in the manuscript:

- Page 20, Line 433: Statistical analyses: strategy and approach

A first-order reaction equation is built to represent the release kinetics of additives from plastic particles to the water phase. Let X (ngg^{-1}) be a chemical quantity released in a given medium. The dynamics of X is given with:

$$X(t) = a(1 - \exp(-bt)),$$

where a and b are parameters that must be estimated from the data. Note that $X(t)$ is a strictly concave increasing function that goes to a (ngg^{-1}) as time t (d) goes to infinity and where b ($\text{ngg}^{-1}\text{d}^{-1}$) is the rate at which $X(t)$ goes to its plateau. Starting with experimental data (t_i, x_i) , $i = 1, \dots, n$, parameters a and b are estimated by non-linear regression (Gauss-Newton descent-gradient algorithm)⁵⁷ when minimizing the cost function:

$$\text{SSE}(a, b) = \sum_{i=1}^n [X(t_i) - x_i]^2,$$

which measures the distance from the data to the model, where $X(t_i)$ is the value of the model at time t_i . Initial conditions for parameter estimation are chosen empirically so that the first fit is reasonable. Once

estimations (\hat{a}, \hat{b}) have been found, a confidence region can be designed as a part of the parameter space for which boundaries are defined with:

$$\left\{ (a, b): \text{SSE}(a, b) \leq \left(1 + \frac{p}{n-p} F_{p, n-p}^{\alpha} \right) \text{SSE}(\hat{a}, \hat{b}) \right\},$$

where p is the number of parameters, n is the number of observations and $F_{p, n-p}^{\alpha}$ is the α -order quantile of the Fisher–Snedecor distribution with $(p, n - p)$ degrees of freedom. For a large number of observations (n), this region will have the required asymptotic confidence level $100(1 - \alpha)\%$ (see Seber, 2003 for more details)⁵⁷ and is also reliable for finite n . The kinetics curve can be estimated using pairwise observations (a, b) randomly drawn inside the confidence region. This makes the confidence bands such as those presented in Fig. 1 and Supplementary Figure 1. In this way, it is possible to compare dynamics between compounds or treatments all along the time course with statistical significance. The more the dynamics are different, the less their 95% confidence intervals overlap.

In addition, a principal component analysis PCA was run. A 2D-mapping explained 90% of the entire variability and showed 2 groups of correlated compounds used to justify the choice of DMP and TEHP as model compounds:

- Page 7, Line 134: This selection was driven by a principal component analysis performed on all additive release data, showing 2 groups on the second axis (Supplementary Figure 3).
- Supplementary Figure 3. Principal Component Analysis (PCA) performed on all additive release data:

⇒ *“I strongly disagree and suggest that none of the conclusions of their study are supported by the results, since statistical analysis, as they state “did not show any significant differences between the treatments.””*

We hope the clarifications outlined above, together with the in-depth rethinking and depiction of the statistical strategy, are sufficient to convince the editor and the reviewer of the soundness of our results and the appropriateness of our conclusions.

REVIEWERS' COMMENTS

Reviewer #3 (Remarks to the Author):

I served as Reviewer 3 on the previous submission of this manuscript. The authors have effectively responded to my comments. I have no further concerns regarding this manuscript.

Manuscript NCOMMS-19-39508-C

N.B.: all authors inputs and answers are noted in blue in this document and the associated revised manuscript.

Reviewer #3 (Remarks to the Author):

I served as Reviewer 3 on the previous submission of this manuscript. The authors have effectively responded to my comments. I have no further concerns regarding this manuscript.

The co-authors would like to deeply thank R3 for taking the time to review our revised manuscript and we are grateful for the reviewer's comments to previous revisions that have enabled us to strengthen the manuscript on the statistical side.